# Therapeutic Efficacy of *Helianthemum lippii* Extract and Silver Nanoparticles Synthesized from the Extract against Cadmium-Induced Renal Nephrotoxicity in Wistar Rats

**DOI:** 10.3390/ph17080982

**Published:** 2024-07-25

**Authors:** Ibtissam Laib, Boutlilis Djahra Ali, Ali Alsalme, David Cornu, Mikhael Bechelany, Ahmed Barhoum

**Affiliations:** 1Department of Cellular and Molecular Biology, Faculty of Natural and Life Sciences, El Oued University, El Oued 39000, Algeria; laib-ibtissam@univ-eloued.dz (I.L.); djahra@yahoo.fr (B.D.A.); 2Laboratory of Biology, Environment and Health, Faculty of Natural and Life Sciences, El Oued University, El-Oued 39000, Algeria; 3Higher School of Saharan Agriculture, El Oued 39000, Algeria; 4Department of Chemistry, College of Science, King Saud University, Riyadh 11451, Saudi Arabia; aalsalme@ksu.edu.sa; 5Institut Européen des Membranes (IEM), UMR 5635, University of Montpellier, ENSCM, CNRS, 34095 Montpellier, France; david.cornu@umontpellier.fr (D.C.); mikhael.bechelany@umontpellier.fr (M.B.); 6Functional Materials Group, Gulf University for Science and Technology (GUST), Mubarak Al-Abdullah 32093, Kuwait; 7NanoStruc Research Group, Chemistry Department, Faculty of Science, Helwan University, Cairo 11795, Egypt

**Keywords:** *Helianthemum lippii*, silver nanoparticles, cadmium toxicity, nephrotoxicity, biochemical parameters, oxidative stress, histopathology, Wistar rats, intraperitoneal injection, gavage

## Abstract

This study explored the therapeutic efficacy of *Helianthemum lippii* and silver nanoparticles (Ag NPs) synthesized using a *H. lippii* extract to alleviate cadmium-induced nephrotoxicity in Wistar rats. Sub-acute toxicity assessments of *H. lippii* (100 mg/kg, 1000 mg/kg, and 4000 mg/kg) and Ag NPs (2 mg/kg and 10 mg/kg) did not find any significant difference, compared with untreated control rats (*n* = 3 animals/group). Then, the adult Wistar rats were divided into one control (untreated/unexposed) and six experimental groups (*n* = 5/group): Ag NPs alone, *H. lippii* alone, exposure to 50 mg/kg CdCl_2_ in drinking water for 35 days, exposure to CdCl_2_ for 35 days followed by treatment with 0.1 mg/kg/day Ag NPs (intraperitoneal injection) and/or 100 mg/kg/day *H. lippii* by gavage for 15 days. In the CdCl_2_-exposed group, body weight decreased; urea, creatinine, and uric acid concentrations increased (*p* < 0.05 vs. control), indicative of nephrotoxicity, antioxidant defenses (SOD, GSH, and CAT) were reduced, and malondialdehyde concentration increased. Moreover, the kidney’s architecture in CdCl_2_-exposed rats was altered: fibrosis, inflammatory cell infiltration, glomerular destruction, and tubular dilatation. Treatment with *H. lippii* and/or Ag NPs after CdCl_2_ exposure improved some of the renal function and architecture alterations induced by CdCl_2_, and also increased body weight. This study underscores the potential therapeutic applications of *H. lippii* and Ag NPs to decrease oxidative stress and promote xenobiotic detoxification, in line with the growing emphasis on environmentally conscious practices in scientific research and healthcare.

## 1. Introduction

Heavy metal contamination in food and feedstuffs poses significant risks to human and animal health, making it a global issue [1]. Cadmium (Cd), a naturally occurring heavy metal with high bioaccumulation potential, is released into the environment, particularly in industrialized areas. Cd is lethal even at extremely low doses. Chronic exposure to low Cd concentrations causes adverse health effects in pediatric and adult populations, including reduced glomerular filtration rate and fecundity [2,3]. The tolerable Cd intake is 0.83 µg/kg body weight/day, and a urinary Cd excretion rate of 5.24 µg/g creatinine is considered the threshold level for toxicity [2]. However, recent studies suggest that these levels may not provide sufficient health protection because adverse health effects are observed at lower Cd accumulation levels. Cadmium chloride (CdCl_2_) targets various organs, including the bones, lungs, skeletal muscles, heart, and kidneys [4]. Cd toxicity is due to its promotion of reactive oxygen species (ROS) production by reducing the levels of antioxidant enzymes, such as catalase, superoxide dismutase (SOD), and glutathione (GSH) [5]. Chronic Cd exposure often leads to renal damage, resulting in kidney disease. Various pathways contribute to organ damage induced by cadmium exposure, including cell apoptosis, autophagy [6], necrosis, cell–cell junction disruption [7], and cell-signaling pathway disorganization. Cd impact on membrane-dependent functions is often attributed to the induction of lipid peroxidation, which affects many enzymatic activities, possibly through the displacement of beneficial metals from active sites or direct binding to the enzyme-active sites [8]. In kidney tissue, Cd accumulates prominently in the proximal convoluted tubules, leading to glycosuria, proteinuria, and aminoaciduria [9,10].

In recent years, interest has been increasing in alternative and safer therapeutic methods to address Cd-induced renal damage due to the considerable health risks associated with heavy metal toxicity [11]. Previous studies highlighted the susceptibility of vital organs, such as the liver and kidneys, to Cd toxicity [12]. Conventional strategies have explored the use of metals (e.g., zinc, copper, and selenium) as antagonists or antioxidants to mitigate Cd toxicity; however, they must be administered at high doses, and this can pose serious risks [13], such as gastrointestinal disturbances, neurotoxicity, and imbalances in essential minerals within the body. Moreover, the efficacy of these treatments can be inconsistent, sometimes offering only partial protection against Cd-induced damage, which leaves patients at ongoing risk of health complications. Consequently, attention has shifted to plant-based remedies, and many studies have investigated the potential of plant extracts to provide safe and effective protection against Cd-induced nephrotoxicity [14]. The strong antioxidant and protective activities of plant extracts in tissues and membrane lipids are mediated through free radical scavenging in a dose-dependent manner. Therefore, plant extracts have been tested in various experimental models as protective agents against various toxicants [15]. Due to their antioxidant properties and minimal side effects, plant (e.g., green tea, curcumin, and black cumin) extracts have emerged as promising alternatives for managing Cd-induced kidney damage [16,17,18]. This shift towards plant-based compounds underscores the importance of exploring nature-derived solutions to mitigate the adverse effects of heavy metal exposure on human health.

Simultaneously, nanoparticle-based therapies present innovative solutions for the targeted and effective treatment of heavy-metal-induced nephrotoxicity. Nanoparticles (NPs) can be engineered to deliver therapeutic agents directly to the kidneys, thereby enhancing treatment efficacy while minimizing systemic side effects [19]. Among NPs, silver nanoparticles (Ag NPs) have gained prominence due to their unique physicochemical properties, including a high surface area-to-volume ratio, excellent conductivity, and anti-inflammatory activity, making them versatile for various applications, including heavy metal detoxification [20,21]. Ag NPs synthesized using plant extracts exhibit promising capabilities for targeted antioxidant and anti-inflammatory effects. They also demonstrate a strong affinity for heavy metals like cadmium, effectively binding and sequestering cadmium ions [22]. This dual action of sequestration and antioxidant activity helps mitigate oxidative stress and inflammation in the affected tissues. Furthermore, NPs can enhance the bioavailability of phytochemicals, ensuring higher concentrations reach the affected tissues to exert protective effects, underscoring their potential as therapeutic agents in heavy metal detoxification efforts.

*Helianthemum lippii* is recognized for its abundance of phytochemicals [23,24]. It presents numerous therapeutic advantages and serves as a promising candidate for nanoparticle (NP) synthesis [21]. Its phytochemical constituents, such as flavonoids, phenolic acids, tannins, and alkaloids [25], are renowned for their robust antioxidant, anti-inflammatory, anti-ulcer, analgesic, anticancer, anti-biofilm, and antimicrobial properties [26]. These compounds are crucial in mitigating oxidative stress, scavenging free radicals, and modulating signaling pathways implicated in various diseases, particularly in heavy-metal-induced nephrotoxicity. Moreover, *H. lippii* phytochemicals are adept in the synthesis of zinc NPs [27] and silver (Ag) NPs [21], where they serve as effective reducing and stabilizing agents in green synthesis processes, ensuring biocompatibility and therapeutic efficacy. Leveraging *H. lippii* phytochemicals for NP synthesis holds immense promise for therapeutic applications, such as targeted delivery, enhanced bioavailability, and reduced cytotoxicity. The synergistic interactions among these phytochemicals further amplify their therapeutic effects, establishing *H. lippii-*based NPs as a compelling option for addressing a range of ailments, including heavy-metal-induced nephrotoxicity.

Several notable studies have demonstrated the efficacy of phytochemicals and NPs for therapeutic applications, affirming the viability of this approach. Recent research on green-synthesized Ag NPs using plant extracts from *Azadirachta indica* and *Moringa oleifera* has shown significant antioxidant and antimicrobial activities, showcasing the potential of phytochemical-mediated NP synthesis [28]. Moreover, investigations into plant-based NPs derived from *Camellia sinensis* (green tea) and aloe vera have revealed promising results in managing oxidative stress and inflammation across various disease models [29,30]. These findings, coupled with the current study, underscore the potential of H. lipid-based NPs in providing effective and environmentally conscious solutions for managing heavy-metal-induced nephrotoxicity. This study explored the therapeutic potential of *H. lippii* and Ag NPs, synthesized using a *H. lippii* extract, for alleviating Cd-induced nephrotoxicity in Wistar rats. Initial sub-acute toxicity assessments of various doses of *H. lippii* and Ag NPs indicated their safety, compared to untreated controls. The adult Wistar rats were then divided into seven groups, including untreated control and experimental groups treated with *H. lippii* or Ag NPs alone, exposed to CdCl_2_, or exposed to CdCl_2_ followed by treatment with *H. lippii* and/or Ag NPs. Monitoring of body weight, renal function markers, and oxidative stress indicators provided insights into the nephrotoxic effects of CdCl_2_ exposure and the potential protective effects of *H. lippii* and Ag NPs. A histological examination revealed CdCl_2_-induced alterations in the kidney architecture, which were mitigated by treatment with *H. lippii* and Ag NPs, highlighting their promising therapeutic applications in managing oxidative stress and xenobiotic detoxification, and emphasizing environmentally conscious practices in scientific research and healthcare.

## 2. Results and Discussion

### 2.1. Phytochemical Screening

Table 1 summarizes the results of the phytochemical analysis of the *H. lippii* aqueous extract. The extract contained polyphenols, catechin and gallic tannins, flavonoids, saponins, anthocyanins, leucoanthocyanins, cardiac glycosides, steroids, terpenoids, and mucilages. Alkaloids were absent when tested with a Mayer reagent, but present according to a Wagner test. These findings highlight the rich diversity of phytochemicals present in the *H. lippii* aqueous extract, suggesting its potential therapeutic value. They also underscore the plant’s importance in traditional medicine and its potential applications in pharmaceutical and nutraceutical industries. More studies are needed to explore the specific health benefits and pharmacological activities associated with these different phytochemicals.

### 2.2. HPLC-Based Quantitative Analysis

An HPLC analysis of the *H. lippii* aqueous extract (Figure 1 and Table 2) allowed for the identification of only six phenolic compounds, despite the presence of many peaks. Gallic acid was the most abundant, followed by chlorogenic acid, quercetin, naringin, p-coumaric acid, and caffeic acid. These findings provide valuable insights into the chemical composition of the *H. lippii* extract and its potential therapeutic applications. The significant presence of gallic acid, chlorogenic acid, and quercetin suggests their potential contribution to the medicinal properties attributed to *H. lippii*. Previous studies highlighted the antioxidant and anticancer properties of gallic acid and catechin [30]. Caffeic acid exhibits antioxidant and iron-chelating activities and reduces acute immune and inflammatory responses [31]. Naringin and quercetin, present at moderate levels in the extract, are potent antioxidants with promising activity against oxidative damage [32]. The existence of these secondary metabolites in the *H. lippii* extract aligns with its traditional use as a remedy for various diseases, providing scientific evidence to support its pharmacological potential. The identification and quantification of these phenolic compounds enrich our understanding of *H. lippii* therapeutic properties and pave the way for future research for medicinal applications. Future studies may elucidate the specific mechanisms of action and the therapeutic efficacy of the *H. lippii* extract’s components. However, in the HPLC chromatogram (Figure 1), notable peaks were observed at 10–12 min. These peaks were not identified in the present study but suggest the presence of other significant phytochemicals in the *H. lippii* extract. Future research will identify these compounds using more advanced analytical techniques and additional standards.

### 2.3. Characteristics of the Ag NPs Fabricated Using the H. lippii Extract

The XRD pattern of the synthesized Ag NPs included distinct diffraction peaks at 38.2°, 44.4°, 64.7°, and 77.5°, corresponding to the (111), (200), (220), and (311) crystallographic planes of the face-centered cubic unit cell, respectively. These peaks were in excellent agreement with the characteristic peaks of metallic silver (JCPDS Card No. 04-0783) [33]. The well-defined features of these peaks indicated the remarkable crystallinity of the Ag NPs. The Scherrer equation and the peak full width at half maximum yielded a crystallite size of 12.84 nm, confirming the nano-scale dimensions and effectiveness of the green synthesis process (Figure 2a). Similar observations were previously reported by other researchers who studied various plant-based synthesis methods. Figure 2c displays particle size ranges from 10 to 90 nm, with an average particle size distribution of 35 nm, highlighting their uniformity and stability (Figure 2b,c). Meanwhile, an SEM analysis revealed the Ag NPs spherical morphology. The Ag NPs elemental composition was confirmed by the SEM-energy-dispersive X-ray spectroscopy analysis that displayed a prominent signal in the silver region and additional peaks corresponding to carbon and oxygen. These additional peaks suggest the presence of residual components from the synthesis process or potential stabilizing agents. These findings are in agreement with the results reported by (Barhoum et al., 2015 [33]; Ghasemi et al., 2024 [34]; Khanal et al., 2022 [35]; Asefian et al., 2024 [36]).

### 2.4. Sub-Acute Toxicity Study

The sub-acute toxicity test data of the *H. lippii* aqueous extract and Ag NPs in Wistar albino rats indicate no significant adverse effects across all parameters and doses tested. The rats, observed at various intervals (3 h, 24 h, 7 days, and 14 days) after administration, showed no abnormalities in body weight changes, movement, or eye condition and none experienced diarrhea or death (Table 3). Both the control groups and the low-, medium-, and high-dose groups of the *H. lippii* extract (100 mg/kg, 1000 mg/kg, and 4000 mg/kg) and Ag NPs (2 mg/kg and 10 mg/kg) maintained normal conditions throughout the study, suggesting that neither *H. lippii* extract nor Ag NPs exhibit sub-acute toxicity at these doses. This lack of toxicity, even at relatively high doses, is a promising indicator for their potential use as therapeutic agents. Given the observed safety profile, *H. lippii* aqueous extract and Ag NPs have the potential to be developed as therapeutic drugs, specifically for combating Cd-induced renal nephrotoxicity. Cd-induced nephrotoxicity is a serious health concern, and the development of effective treatments with minimal side effects is crucial. The normal health parameters observed in the rats suggest that these compounds could be administered safely without causing harm, a critical consideration in drug development. Further studies, including comprehensive analyses of kidney function parameters, oxidative stress markers, histopathological features, body weight changes, and renal weight changes after exposure to CdCl_2_ and treatment with *H. lippii* extract and Ag NPs, were conducted and are discussed in the next sections to evaluate the therapeutic efficacy of *H. lippii* extract and Ag NPs in mitigating renal damage caused by cadmium.

### 2.5. H. lippii and Ag NP Effect on Cadmium-Induced Nephrotoxicity

Then, the effects of the *H. lippii* extract and Ag NPs on Cd-induced kidney toxicity were investigated in the adult Wistar albino rats (*n* = 5 animals/group) exposed or not to CdCl_2_ (50 mg/kg body weight/day) in drinking water for 35 days. Then, the rats that were treated or not treated with the Ag NPs (0.1 mg/kg, body weight/day by intraperitoneal injection) and/or *H. lippii* extract (100 mg/kg, body weight/day by gavage) were investigated for 15 days. Some mice received only the Ag NPs (0.1 mg/kg, body weight/day by intraperitoneal injection) or the *H. lippii* extract (100 mg/kg, body weight/day by gavage) for 35 days.

#### Body Weight and Relative Kidney Weight

An analysis of the body weight changes and relative kidney weight at the end of the study revealed important insights into the impact of CdCl_2_ exposure and the potential protective effects of *H. lippii* and Ag NPs. In the control group (Group 1; baseline body weight = 282 ± 14.7 g), body weight increased by 0.072 ± 0.02 g/day (Figure 3a,b). Conversely, upon exposure to CdCl_2_ (Group 2; (baseline body weight = 244.8 ± 2.69 g), body weight decreased by −0.181 ± 0.06 g/day. According to (Poli et al., 2022) [37], this may be explained by increased lipid degeneration caused by Cd toxicity [38], resulting in a significant decrease in food uptake, food avoidance, or post-food palatability due to Cd toxicity. Moreover, the activation of oxidative stress results in changes in antioxidant status and leads to severe metabolic abnormalities and weight loss [39]. Conversely, exposure to *H. lippii* alone (Group 3; baseline body weight = 244.8 ± 2.69 g) or Ag NPs alone (Group 5; baseline body weight = 231.8 ± 15.7 g) did not affect the normal pattern of body weight gain, as indicated by their weight increase by 0.119 ± 0.09 g/day and 0.042 ± 0.01 g/day, respectively, further confirming their relative safety. The administration of *H. lippii* or Ag NPs after CdCl_2_ exposure (Group 4 and Group 6 with baseline body weight = 247.2 ± 1.1 g and 247.2 ± 1.1 g, respectively) limited body weight loss (−0.05 ± 0.2 g/day and 0.07 ± 0.009 g/day, respectively). Moreover, in Group 7 (exposure to CdCl_2_ followed by administration of both *H. lippii* and Ag NPs; baseline body weight = 234.0 ± 13.0 g), the body weight increase was comparable to that in the control group (0.072 ± 0.02 g/day). This suggests a potential protective role of the *H. lippii* extract and Ag NPs against Cd-induced metabolic abnormalities and oxidative stress. These findings underscore the therapeutic potential of *H. lippii* and Ag NPs in mitigating Cd-induced toxicity.

The comparison of the relative kidney weight in the different groups at the end of the study (Figure 3c) showed that, compared to Group 1 (control), it was significantly increased in Group 2 (exposure to CdCl_2_) (0.46 ± 0.04 vs. 0.55 ± 0.02), suggesting a hypertrophic response induced by Cd exposure. Conversely, in Group 3 (*H. lippii* alone) and Group 5 (Ag NPs alone), the relative kidney weights were 0.43 ± 0.03 and 0.42 ± 0.02, respectively, reaffirming their safety profiles (Figure 3c). In Group 4 (CdCl_2_ + *H. lippii* extract), Group 6 (CdCl_2_ + Ag NPs), and Group 7 (CdCl_2_+ *H. lippii* +Ag), the relative kidney weights were significantly decreased, compared with Group 2: 0.48 ± 0.01, 0.48 ± 0.02, and 0.47 ± 0.01, respectively. This suggests a potential reversal of Cd-induced hypertrophy, underscoring the protective effects of the *H. lippii* extract and Ag NPs against Cd toxicity. These results are in agreement with the findings by (Liu et al. 2022) [40], who tested Ag NPs as nanostructures for novel and improved biological uses.

### 2.6. Biochemical Parameters

Figure 4 summarizes the data on the effect of CdCl_2_ exposure and treatment with the *H. lippii* extract and/or Ag NPs on different serum markers (Appendix A). In Group 2 (CdCl_2_ exposure), serum triglycerides (0.79 ± 0.01 g/L) (Figure 4a), cholesterol (0.78 ± 0.02 mg/L) (Figure 4b), and glucose (1.13 ± 0.09 g/L) (Figure 4c) were significantly increased, compared with Group 1 (control). These findings indicate disrupted lipid metabolism and suggest potential liver damage affecting protein synthesis, as evidenced by the reduction in the serum albumin concentration (29.5 ± 0.0131 g/L) (Figure 4d). Conversely, these markers were not significantly different in Group 3 (*H. lippii* extract alone) and Group 5 (Ag NPs alone), compared with the control group, indicating that the *H. lippii* extract and Ag NPs alone do not alter the lipid profiles, glucose metabolism, or albumin levels. In Group 4 (*H. lippii* extract after CdCl_2_ exposure), Group 6 (CdCl_2_ exposure + Ag NPs), and Group 7 (CdCl_2_ exposure + *H. lippii* extract + Ag NPs), these parameters tended to be normalized, suggesting a mitigating effect on CdCl_2_-induced alterations. Specifically, serum triglycerides, cholesterol, glucose, and albumin levels were within normal ranges, according to controls (triglycerides: 0.63 ± 0.01 g/L, 0.59 ± 0.05 g/L, 0.40 ± 0.01 g/L; cholesterol: 0.75 ± 0.01 mg/L, 0.63 ± 0.03 mg/L, 0.75 ± 0.01 mg/L; glucose: 0.82 ± 0.10 g/L, 1.08 ± 0.009 g/L, 0.88 ± 0.06 g/L; albumin: 31.33 ± 0.44 g/L, 29.66 ± 0.4 g/L, 30.75 ± 0.3 g/L, respectively).

Most biochemical markers in the groups treated with the *H. lippii* extract and/or Ag NPs were restored to levels near those of the control group, likely due to the anti-inflammatory and antioxidant properties of *H. lippii*. The tannic acid in *H. lippii* enhances glucose uptake by tissues, suppresses gluconeogenesis, and activates insulin secretion from β-cells [41], while naringenin protects the islets of Langerhans from degenerative changes. In rats exposed to cadmium and treated with *H. lippii* and/or Ag NPs, lipid parameters, serum albumin levels, protein levels, and calcium concentrations were regulated, attributed to the presence of flavonoids and phenolic compounds in the extract that scavenge superoxide, hydroxyl ions, and peroxyl radicals, preventing lipid peroxidation, lowering lipogenic enzyme activity, and enhancing lipoprotein lipase function [42]. These flavonoids’ antioxidant capacities also positively affect protein levels and calcium. Additionally, silver nanoparticles improve lipid profiles, protein levels, and blood glucose levels, consistent with their anti-hyperglycemic, antioxidant, and anti-inflammatory responses [43,44]. The combined treatment of *H. lippii* and Ag NPs significantly restored most biochemical parameters, suggesting that the antioxidants in both treatments synergistically improved these levels against CdCl_2_ exposure. Ag-based nanomaterials are less toxic to biological systems and have greater therapeutic efficacy when bio-conjugated with plant extracts due to the variety of phytochemicals or capping agents provided by the plant extracts, combined with the small size and large surface area of Ag NPs, which allows for maximum adsorption of these capping agents onto their surfaces. This combined treatment offers a promising strategy for addressing renal abnormalities caused by heavy-metal toxicity [45,46].

These findings underscore the potential therapeutic role of the *H. lippii* extract and Ag NPs in ameliorating CdCl_2_-induced effects on lipid profiles, glucose metabolism, and protein synthesis, warranting further investigations into their clinical applications in the context of Cd toxicity. Similarly, (Rehman et al., 2023) and El-Baz et al., (2023) [47,48] previously showed Ag NP’s restorative activity and ability to improve lipid profiles, protein levels, and blood glucose levels, in line with their anti-hyperglycemic, antioxidant, and inflammatory responses.

### 2.7. Biochemical Biomarkers of Renal Function

Figure 5 summarizes renal function biomarkers across the seven experimental groups (Appendix A). In Group 2 (CdCl_2_-treated), the serum urea (0.58 ± 0.02 g/L), creatinine (5.36 ± 0.001 mg/L), uric acid (20.00 ± 0.001 mg/L), and urea-to-creatinine ratio (108.0 ± 11.2) were significantly elevated, compared to Group 1 (control), indicating CdCl_2_-induced nephrotoxicity, consistent with the findings of (Liu et al., 1998) [49]. Conversely, Group 3 (*H. lippii* extract) and Group 5 (Ag NPs) showed values similar to controls: urea (0.45 ± 0.022 g/L, 0.46 ± 0.025 g/L), creatinine (4.81 ± 0.05 mg/L, 4.010 ± 0.0001 mg/L), uric acid (11.33 ± 1.45 mg/L, 11.67 ± 1.28 mg/L), and urea-to-creatinine ratio (94.87 ± 6.7, 104.74 ± 8.60), indicating biocompatibility under normal conditions. Group 4 (CdCl_2_ + *H. lippii* extract) showed decreased levels, compared to Group 2: urea (0.47 ± 0.028 g/L), creatinine (4.25 ± 0.001 mg/L), uric acid (16.25 ± 1.50 mg/L), and urea-to-creatinine ratio (110.9 ± 10.0), suggesting that the *H. lippii* extract mitigates CdCl_2_-induced renal damage via antioxidant compounds. Similarly, Group 6 (CdCl_2_ + Ag NPs) displayed reduced biomarker levels: urea (0.47 ± 0.01 g/L), creatinine (5.11 ± 0.08 mg/L), uric acid (12.5 ± 0.77 mg/L), and urea-to-creatinine ratio (95.25 ± 6.46), indicating Ag NPs alleviate CdCl_2_-induced renal damage. Group 7 (CdCl_2_ + *H. lippii* extract + Ag NPs) exhibited comparable trends: urea (0.48 ± 0.02 g/L), creatinine (4.81 ± 0.08 mg/L), uric acid (11.75 ± 0.55 mg/L), and urea to creatinine ratio (95.1 ± 11.4), highlighting combined effects in restoring renal function post CdCl_2_-induced nephrotoxicity.

The protective effect of Ag NPs was evident in the significant reduction in kidney function biomarkers, compared to the Cd-exposed group, consistent with findings by (Rehman et al., 2023) [47]. Ag NPs stabilize membranes, combat oxidative stress, and mitigate inflammatory responses, thereby limiting intracellular enzyme leakage. Studies underscore their cytoprotective activities both in vitro and in vivo [48]. The antioxidant- and free-radical-scavenging properties of Ag NPs likely contribute to restoring biochemical and histological parameters close to normal levels, highlighting their potential therapeutic value for kidney function protection. Group 4 (Cd exposure followed by *H. lippii*) exhibited improved kidney biomarkers due to the extract’s high concentration of bioactive compounds (gallic acid, chlorogenic acid, caffeic acid, p-coumaric acid), which protect against cellular injury. Furthermore, *H. lippii*’s bioactive compounds possess potent anti-inflammatory properties by inhibiting cytokine production (TNF-α, IL-1β, IL-6) [48,49,50,51,52], thereby reducing inflammation and supporting renal tissue repair. These findings align with the previous research demonstrating their antioxidant efficacy against Cd-induced cell damage [48,49,50,51,52]. Combining *H. lippii* extract with silver nanoparticles enhanced kidney function, compared to CdCl_2_ alone, illustrating the synergistic renal protective effects of silver compounds and the antioxidant properties of the plant extract.

### 2.8. Oxidative Stress Markers

The analysis of the kidney oxidative stress markers in the seven experimental groups (Figure 6) revealed a significant increase in the MDA concentration in CdCl_2_-exposed rats (Group 2), compared with controls (Group 1) (0.9061 ± 0.0592 vs. 0.403 ± 0.021 nmol/mg protein) (Figure 6a), which is indicative of higher oxidative stress and potential kidney damage (Appendix A). Conversely, antioxidant defense markers were decreased in Group 2, compared with Group 1: SOD activity (0.239 ± 0.00001 vs. 0.26 ± 0.009 U/mg protein) (Figure 6b), GSH levels (0.00022 ± 0.000001 vs. 0.00023 ± 0.001 nmol/g tissue) (Figure 6c), and catalase activity (0.01931 ± 0.0072 vs. 0.04373 ± 0.008 U/g tissue) (Figure 6d). The GSH and catalase are important indicators of oxidative damage to macromolecules, and our findings are compatible with previous findings [53]. (Ijaz et al., 2023) [11] and Albasher et al., (2020) [54] showed that in rats, exposure to Cd in drinking water for 30 days significantly increases the kidney’s MDA concentration and significantly decreases catalase and SOD activities, compared with the control group.

Conversely, these oxidative stress markers did not significantly differ in rats treated only with *H. lippii* extract (Group 3) or Ag NPs (Group 5) compared with controls (Group 1), suggesting no adverse effects on kidney oxidative stress from these treatments alone. Treatment with *H. lippii* following CdCl_2_ exposure (Group 4) resulted in a reduction in the oxidative stress markers and an increase in antioxidant defenses, compared with Group 2. Specifically, the MDA concentration decreased (0.7469 ± 0.0545 nmol/mg protein), while SOD, GSH, and catalase levels increased, indicating an amelioration of kidney oxidative damage. This beneficial effect can be attributed to the high concentration of bioactive compounds in *H. lippii*, such as gallic acid, chlorogenic acid, caffeic acid, and p-coumaric acid, which are known for their potent antioxidant properties [24,55]. *H. lippii* enhances the kidney’s antioxidant defense system by scavenging ROS directly and enhancing the activity of endogenous antioxidant enzymes. These mechanisms play crucial roles in neutralizing oxidative stress and protecting renal cells from damage induced by Cd exposure. Moreover, *H. lippii* facilitates the synthesis of metallothionein, a metal-binding protein that aids in cadmium detoxification and reduces its accumulation in renal tissues, thereby mitigating oxidative-stress-related cellular damage associated with heavy metal toxicity [54,55,56,57].

The rats treated with Ag NPs after CdCl_2_ exposure (Group 6) showed notable reductions in oxidative stress markers and enhanced antioxidant defenses compared to Group 2. Specifically, the MDA concentration decreased (0.4919 ± 0.0594 nmol/mg protein), while the SOD, GSH, and catalase levels increased, indicating that Ag NPs may alleviate kidney oxidative damage through their robust antioxidant and anti-inflammatory properties. Ag NPs exert their protective effects against cadmium toxicity primarily through their potent antioxidant activity [58]. Cadmium exposure triggers the generation of reactive oxygen species (ROS), leading to oxidative stress and renal cell damage [22]. Ag NPs directly scavenge these ROS, thereby mitigating oxidative damage to cellular components such as lipids, proteins, and DNA. Moreover, Ag NPs enhance the activity of endogenous antioxidant enzymes, including SOD, catalase, and GPx [58], and they have been shown to upregulate the expression of Nrf2, a key regulator of cellular defense against oxidative stress, which aids in preventing cadmium-induced cellular damage.

Furthermore, the combined treatment with the *H. lippii* extract and Ag NPs after CdCl_2_ exposure (Group 7) further enhanced protection against oxidative stress. The MDA concentration was reduced (0.5274 ± 0.0382 nmol/mg protein), while SOD, GSH, and catalase levels were elevated compared to Group 2, underscoring the distinct antioxidant efficacy of each component. These results highlight the synergistic benefits of combining plant extracts and Ag NPs in safeguarding kidney structure and function from CdCl_2_-induced toxicity. This combined approach capitalizes on the strengths of natural and nanotechnology-derived therapeutic strategies, enhancing bioavailability, improving drug delivery, boosting therapeutic efficacy, minimizing side effects, offering multifaceted therapy, and potentially overcoming drug resistance. The synergistic therapeutic effects observed underscore the promise of this approach for effectively and safely managing a broad spectrum of diseases.

### 2.9. Histological Alterations in Kidney

The semi-quantitative analysis of kidney tissue samples revealed distinct architectural damage among the experimental groups (Figure 7 and Appendix A). In Group 2 (CdCl_2_ exposure), there were severe histopathological changes, including prominent inflammatory cell infiltration (+++), destroyed glomeruli (+++), and tubular dilation (+++), indicating extensive tissue damage induced by cadmium exposure. These findings align with previous studies reporting Cd-induced structural alterations, such as increased mesangial matrix and glomerular swelling [59]. Cadmium toxicity targets the proximal tubular epithelium, sensitizing it to oxidative stress and leading to kidney cell injury [11]. Elevated nitric oxide and ROS production further exacerbate tissue damage and contribute to kidney dysfunction [60].

Groups 3 and 5, treated with *H. lippii* extract or Ag NPs alone, respectively, exhibited no significant histopathological alterations compared to controls (Group 1), indicating the biocompatibility of these treatments under normal conditions. In Group 4 (*H. lippii* extract after CdCl_2_ exposure), although tubular dilation persisted, inflammatory infiltration and glomerular destruction were notably absent compared to Group 2. This suggests that *H. lippii* extract mitigates CdCl_2_-induced histopathological changes by enhancing antioxidant defenses, reducing tissue inflammation, and modulating the NRF_2_ transcription factor, crucial for antioxidant protein induction [61]. The extract’s flavonoids and phenolic acids scavenge ROS, minimizing lipid peroxidation and preserving renal architecture by attenuating tubular degeneration and interstitial fibrosis [62,63,64]. Additionally, *H. lippii*’s anti-inflammatory properties suppress cytokine production, further protecting renal tissues from inflammatory damage.

In Group 6 (Ag NPs after CdCl_2_ exposure), there were no signs of inflammatory infiltration, tubular dilation, or destroyed glomeruli observed, indicating a protective effect against cadmium-induced histological alterations. Ag NPs exert multifaceted protective mechanisms, including antioxidant, anti-inflammatory, cytoprotective, and detoxifying actions [65,66]. These nanoparticles mitigate oxidative stress and inflammation, reduce apoptosis, and promote tissue repair and regeneration [67]. Moreover, Ag NPs enhance angiogenesis, improving blood flow to damaged areas and facilitating tissue recovery, thereby preserving normal kidney architecture despite cadmium exposure [62,67,68]. The comprehensive protective effects of Ag NPs underscore their potential therapeutic value in preventing and treating Cd-induced nephrotoxicity.

Finally, in Group 7 (*H. lippii* + Ag NPs after CdCl_2_ exposure), the architectural damage parameters were restored to near-normal levels, compared to Group 2. No inflammatory infiltration, tubular dilation, or destroyed glomeruli were observed, highlighting the synergistic effects of combining *H. lippii* extract with Ag NPs. The antioxidants and anti-inflammatory compounds in *H. lippii* complement the unique properties of Ag NPs, enhancing the overall therapeutic efficacy against cadmium toxicity. This combination strategy may offer a comprehensive defense against Cd-induced cellular damage by leveraging antioxidant pathways from *H. lippii* and multifunctional protective mechanisms from Ag NPs. These findings support the therapeutic potential of combining natural plant extracts with nanotechnology-derived nanoparticles for the effective and safe treatment of kidney damage induced by heavy metals like cadmium.

## 3. Material and Methods 

### 3.1. Chemicals and Reagents

The cadmium chloride (CdCl_2_, 99%), silver nitrate (AgNO_3_, 99.9%), potassium dihydrogen phosphate (KH_2_PO_4_, 99.5%), dibasic potassium phosphate (K_2_HPO_4_, 99.95%*),* ascorbic acid (vitamin C, 99.9%), ethylenediaminetetraacetic acid (EDTA, 99.0%), 2-thiobarbituric acid (TBA, 97.0%), salicylic acid (C_7_H_6_O_3_, 95.5%), 5,5′-dithiobis-2-nitrobenzoic acid (DTNB, 98.28%), 4-nitro blue tetrazolium chloride (NBT, 99.9%), ethanolamine (C_2_H_7_NO, 99%), O-cresolphthalein (C_22_H_18_O_4_, 95%,), 8-hydroxyquinoline (C_9_H_7_NO, 99.99%), Tris pH 7.8 ((HOCH_2_)_3_CNH_2_, 99.9%), α-ketoglutaric acid (C_5_H_6_O_5_, 99.5%), 2-4 dichlorophenol ((CH3)_2_C6H3OH, 99%), sodium hydroxide (NaOH, 99%), picric acid (C_6_H_3_N_3_O_7_, 99.8%), 4–aminophenazone (4-AP) (C_13_H_17_N_3_O, 98%), thiobarbituric acid (C_4_H_4_N_2_O_2_S, 99%), sodium chloride (NaCl, 99%), methionine (C_5_H_11_NO_2_S, 99.3%), riboflavin (C₁₇H₂₀N₄O₆, 98%), hydrogen peroxide (H_2_O_2_, 99.9%), ethanol (CH_3_CH_2_OH, 99.5%), hematoxylin (C_16_H_14_O_6_, 85%), eosin (C_20_H_6_Br_4_Na_2_O_5_, 90%), potassium mercuric iodide (HgI_2_·KI, >98%), ferric chloride (FeCl3, 98%), acetic anhydride ((CH3CO)_2_O, 99%), sulfuric acid (H_2_SO_4_, 98%), ammonia (NH_4_OH, 99%), hydrochloric acid (HCl, 99%), and chloroform (CHCl_3_, 94%) were obtained from Sigma-Aldrich (St. Louis, MO, USA). In March 2022, the aerial parts of *H. lippii* were collected in the Elhamadin region, El-Oued province, Algeria (33° 35′ 00″ N 6° 56′ 33″ E). These specimens were identified by a botany specialist. Following identification, the aerial parts were carefully stored in dry, cool conditions away from light at room temperature.

### 3.2. Preparation the H. lippii Aqueous Extract

The *H. lippii* aerial parts (10 g) were infused in 100 mL of distilled water and left at room temperature in the dark for 24 h. Then, the mixture was filtered using paper filters and the filtrate dried completely at 40 °C. The resulting extract was weighed and stored at 4 °C [69]. 

### 3.3. Phytochemical Screening

The phytochemical screening of the *H. lippii* extracts involved specific qualitative tests to identify its chemical constituents, each using standardized reagents and volumes. Alkaloids were detected using two different reagents: Mayer’s reagent (potassium mercuric iodide), and Wagner’s reagent (iodine in potassium iodide). Each reagent (1 mL) was added separately to 1 mL of the extract. The presence of alkaloids was indicated by the formation of a cream-colored precipitate with the Mayer’s reagent, and a brown or reddish brown precipitate with the Wagner’s reagent [70]. Additionally, tannins were identified by adding 2–3 drops of 5% ferric chloride solution to 1 mL of the extract. The presence of catechic tannins was indicated by a blue–black coloration, while gallic tannins produced a greenish black coloration [71]. To detect terpenes and sterols, 1–2 mL of a Liebermann –Burchard reagent (a mixture of acetic anhydride and concentrated sulfuric acid) was added to 1 mL of the extract, followed by gentle heating. The presence of terpenes and sterols was confirmed by the formation of a blue–green coloration [72]. Saponins were identified by their ability to form stable foam. One milliliter of the extract was mixed with 10 mL of distilled water and shaken vigorously. The formation of stable foam indicated the presence of saponins [73]. Additionally, mucilage was detected by adding 10 mL of absolute ethanol to 1 mL of the extract, resulting in a gelatinous mass [74]. Moreover, polyphenolic compounds were identified by adding 2–3 drops of 5% ferric chloride solution to 1 mL of the extract, with colored complexes indicating their presence [75]. Additionally, flavonoids were detected by adding 1 mL of dilute ammonia solution to 1 mL of the extract, followed by 1 mL of concentrated sulfuric acid. The presence of flavonoids was indicated by the formation of a yellow coloration, which disappeared upon standing [76]. Finally, anthocyanins were detected by adding a small amount of hydrochloric acid followed by a small amount of ammonia. If anthocyanins are present, the color will change, showing red [77]. Each method utilized these reagents and volumes to qualitatively assess the presence or absence of specific phytochemicals in the *H. lippii* extract. A positive sign (+) indicated the presence of the phytochemical, whereas a negative sign (−) indicated its absence.

### 3.4. High-Performance Liquid Chromatography (HPLC) Analysis

The phenolic compounds present in the crude extract were identified using HPLC equipped with a UV-Vis type Shimadzu LC20 AL system. The analytical Shim-pack VP-ODSC18 column (4.6 mm × 250 mm, 5 μm) was coupled with a UV-VIS detector SPD 20A (Shimadzu, Kyoto, Japan) and a universal injector (Hamilton 25 μL). The reverse-phase chromatography was performed with non-polar aliphatic residues and with acetonitrile and 0.1% acetic acid as the mobile phase and gradient elution. The flow rate was set at 1 mL/min, and the injection volume was set at 0.45 µL. The injection volumes for standard and sample were 20 µL, and the monitoring wavelength was 268 nm. To identify and confirm the retention times of the compounds, we used nine standards of known compounds, including caffeic acid (Retention time (min): 16.27), p-Coumaric acid (Retention time (min): 23.81), gallic acid (Retention time (min): 5.29), vanillic acid (Retention time (min): 15.53), chlorogenic acid (Retention time (min): 13.39), naringin (Retention time (min): 34.78), rutin (Retention time (min): 28.37), quercetin (Retention time (min): 45.04), and vanillin (Retention time (min): 21.46). These standards were run under the same HPLC conditions as the sample extracts to establish baseline retention times.

### 3.5. Ag NPs Synthesis and Characterization

For the Ag NP synthesis, 90 mL of a 1 mM AgNO_3_ solution was mixed with 10 mL of the *H. lippii* extract, resulting in the appearance of a brownish tinge, indicative of Ag NP synthesis. Then, the mixture was heated to 60 °C and incubated in the dark for 24 h. Ag NP formation was confirmed by UV-vis analysis, followed by centrifugation to ensure complete separation and washed with ethanol and distilled water. The resulting Ag NPs were carefully collected from the final precipitate and dried [21]. To evaluate the Ag NP’s crystallinity and crystal structure, an X-ray diffraction (XRD) analysis was performed using CuKα radiation (30 kV and 20 mA), with a wavelength of 0.154281 Å and a scanning speed of 0.05°. Scanning electron microscopy (SEM) (Thermo Scientific, Quatro, Thermo Fisher Scientific, Dreieich, Germany) was used to assess the Ag NP size and morphology.

### 3.6. Animal Procurement and Housing

A total of 53 adult male Wistar albino rats, with an average weight of 232.36 ± 3.81 g, were bought from the Pasteur Institute Animal Facility, Algiers, Algeria. Upon arrival, rats were acclimated to the controlled laboratory environment at the Department of Molecular and Cellular Biology, University of El-Oued, Algeria, for 2 weeks. The housing conditions were 19 °C, a relative humidity of 64%, and a 12-h light/12-h dark cycle. Throughout the study, rats had ad libitum access to tap water and standard rat chow. Ethical approval for the animal experiments and protocols was obtained from the Institutional Animal Ethical Committee (IAEC), University of El Oued, Algeria.

### 3.7. Cadmium Toxicity Experiment Design

After acclimatization, the rats were randomly allocated into seven groups (*n* = 5 animals/group). The doses of CdCl_2_, Ag NPs, and *H. lippii* were determined based on previous studies and CdCl_2_ LD_50_ [11,27,78], to ensure optimal exposure and treatment durations. The experimental groups were as follows:

Group 1: control group (normal water throughout the study period).Group 2: exposure to 50 mg/kg body weight/day CdCl_2_ dissolved in drinking water for 5 weeks.Group 3: 100 mg/kg body weight/day *H. lippii* aqueous extract by oral gavage for 5 weeks.Group 4: exposure to CdCl_2_ for 5 weeks, followed by treatment with 100 mg/kg body weight/day *H. lippii* (by oral gavage) for the last 15 days.Group 5: 0.1 mg/kg body weight/day Ag NPs by intraperitoneal injection for 5 weeks.Group 6: CdCl_2_ exposure for 5 weeks, followed by treatment with 0.1 mg/kg body weight/day of Ag NPs (intraperitoneal injection) for the last 15 days.Group 7: CdCl_2_ exposure for 5 weeks and then treatment with 100 mg/kg body weight/day *H. lippii* (oral gavage) and 0.1 mg/kg body weight/day Ag NPs (intraperitoneal injection) for the last 15 days.

The study meticulously monitored the welfare of the rats, with regular recording of their body weights. Established protocols were followed to minimize rat movement and ensure humane procedures. The chloroform was used solely for anesthesia, not euthanasia, in accordance with the intended protocol. A low dose of 94% chloroform in cotton was administered via inhalation to induce dizziness in the rats. Subsequently, euthanasia was conducted via decapitation to ensure swift and humane termination. Blood collection was then performed from the severed neck, adhering to the ethical standards that were thoroughly confirmed [79,80].

### 3.8. Sub-Acute Toxicity Study

The acute toxicity of Ag NPs and *H. lippii* aqueous extract was investigated following the guidelines established by the Organization for Economic Cooperation and Development (OECD) 425. Six groups, each comprising three male Wistar albino rats, were established as follows:

Group 1 (control group): normal water.Groups 2 to 4: administration by oral gavage of one dose of *H. lippii* aqueous extract (100, 1000, 4000 mg/kg)Groups 5 and 6: one intraperitoneal injection of Ag NPs (2, 10 mg/kg).

The sub-acute toxicity was evaluated by monitoring various parameters at different time intervals until day 14: symptoms of toxicity, body weight changes, side effects, movement patterns, diarrhea, ocular abnormalities, and death.

### 3.9. Body Weight

The rats were weighed regularly. The mean body weight was calculated by summing the body weight of each mouse and dividing this number by the number of rats. The body weight changes (%) from baseline were computed using the following formula:Body Weight Change (%) = (Final Body Weight − Baseline Body Weight/Baseline Body Weight) × 100

### 3.10. Relative Kidney Weight

The kidneys were weighed after sacrifice and the relative weight was calculated as follows:Relative kidney weight = (kidney weight/final body weight) × 100

### 3.11. Renal Function Biomarkers

The renal function was evaluated by measuring key serum biomarkers (urea, uric acid, creatinine, glucose, albumin, cholesterol, and triglycerides) using the appropriate Spinreact kits (Barcelona, Spain), following the manufacturer’s instructions.

### 3.12. Oxidative Stress Markers

To determine the malondialdehyde (MDA) level, 200 µL of kidney tissue and 800 µL of the TBA reagent were combined in glass test tubes that were tightly capped and heated at 100 °C in a water bath for 15 min. After 30 min of cooling in a cold-water bath, the tubes were left open to allow the reaction gas escape. The supernatant absorbance at 532 nm was measured using a spectrophotometer after centrifugation at 3000 rpm for 5 min [81].

To determine the GSH level, 200 µL of salicylic acid (0.25%) was mixed with 800 µL of kidney tissue, followed by centrifugation at 1000 rpm for 5 min. Then, 25 µL of DTNB (0.01 mol/L) and 500 µL of the supernatant were added to 1000 µL of tris buffer (0.4 M tris, 0.02M NaCl, pH = 8.9). After 5 min of incubation, the absorbance was measured at 412 nm [82].

To determine the SOD activity, 50 µL of each kidney tissue was mixed with 1000 µL of 0.1 mM EDTA/13 mM methionine, followed by the addition of 1800 µL of 50 mM phosphate buffer, and then 85 µL of NBT and 22.6 µL of riboflavin. Absorption was measured at 560 nm. The same protocol was used for the blank where 50 µL of the phosphate buffer was added instead of the sample [82].

The catalase activity was determined as previously described [40]. The reaction was initiated by combining 200 μL of 0.030 M H_2_O_2_ with 20 μL of each kidney tissue and 780 μL of 0.1 M phosphate buffer (pH 7.5). H_2_O_2_ decomposition was tracked by measuring the decrease in absorbance at 240 nm every 30 s for 2 min. The enzymatic activity was quantified as international units per minute per gram of protein (IU/min/g of protein).

### 3.13. Histopathological Analysis of Kidney Tissues

The kidneys were isolated from the euthanized rats and immediately fixed in a 10% formaldehyde solution. Then, they were dehydrated in ethanol at increasing concentrations (60%, 70%, 80%, and 100%), followed by clearing with xylene and embedding in paraffin. Sections (4~6 µm thickness) were cut from the paraffin-embedded blocks using a Histoline rotary microtome (Thermo Scientific Micron HM 325, Histoline, Milan, Italy). The tissue sections were stained with hematoxylin and eosin and examined under a light microscope to detect morphological changes or abnormalities.

## 4. Statistical Analysis

Data were expressed as mean ± standard error of the mean (SEM). Groups were compared using Student’s *t*-test. Statistical analysis was carried out using Minitab (Version 13Fr) and Excel (version 2007).

## 5. Conclusions

This study investigated the therapeutic potential of *Helianthemum lippii* (*H. lippii)* extract and silver nanoparticles (Ag NPs) synthesized using *H. lippii* extract in mitigating cadmium (Cd)-induced nephrotoxicity in adult Wistar rats (*n* = 5/group). The experimental groups included untreated controls, rats exposed to CdCl_2_ for 35 days, and those treated with *H. lippii* or Ag NPs alone post-Cd exposure, as well as combinations of both treatments. Initial sub-acute toxicity assessments confirmed the safety of *H. lippii* and Ag NPs across various doses, establishing their suitability for therapeutic use. Comprehensive analyses of kidney function parameters, oxidative stress markers, histopathological features, and body weight changes revealed significant improvements following treatment with *H. lippii* and Ag NPs after Cd exposure. Both treatments effectively restored kidney function by reducing creatinine and blood urea nitrogen levels, indicative of improved renal health. Furthermore, *H. lippii* extract and Ag NPs exhibited potent antioxidant properties, enhancing activities of superoxide dismutase (SOD), catalase, and glutathione (GSH), while reducing malondialdehyde (MDA) levels, thus mitigating Cd-induced oxidative stress. Histopathological evaluations demonstrated preserved kidney architecture, reduced inflammatory responses, and minimized tubular dilation and fibrosis in treated groups compared to Cd-exposed controls. Importantly, the combined treatment of the *H. lippii* extract and Ag NPs showed synergistic effects, highlighting enhanced therapeutic efficacy in alleviating Cd-induced nephrotoxicity. These findings underscore the potential of *H. lippii*-based nanoparticles as novel therapeutic agents for managing heavy-metal toxicity, encouraging further research and clinical applications to optimize their use in medical practice.

## Figures and Tables

**Figure 1 pharmaceuticals-17-00982-f001:**
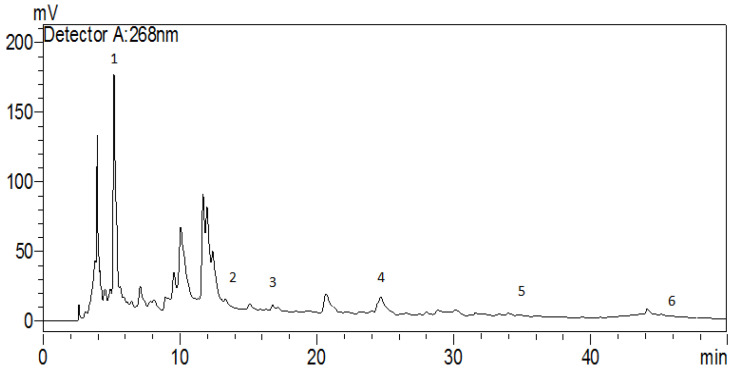
HPLC chromatogram of the *H. lippii* aqueous extract: 1: gallic acid; 2 chlorogenic acid; 3: caffeic acid; 4: p-Coumaric acid; 5: naringin; 6: quercetin.

**Figure 2 pharmaceuticals-17-00982-f002:**
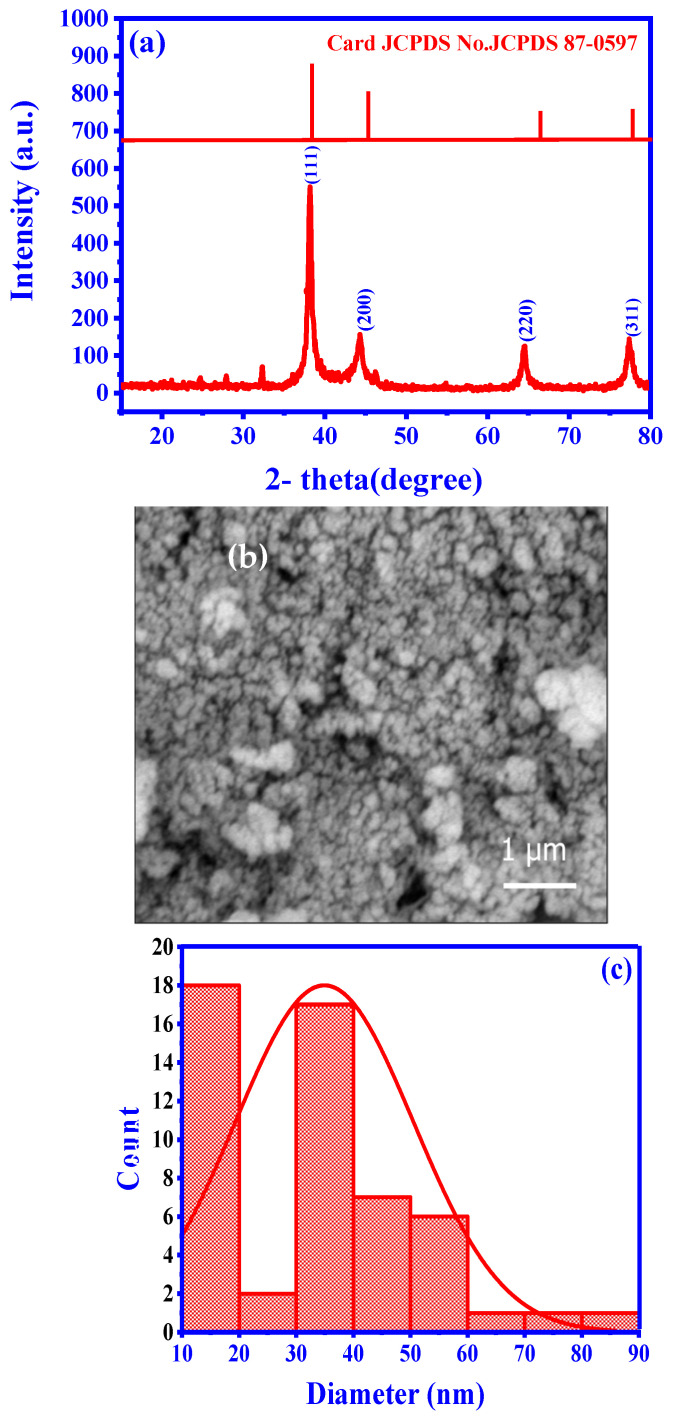
Characteristics of synthesized Ag NPs: (**a**) crystalline structure (XRD data), (**b**) morphology (SEM image), and (**c**) size distribution analysis.

**Figure 3 pharmaceuticals-17-00982-f003:**
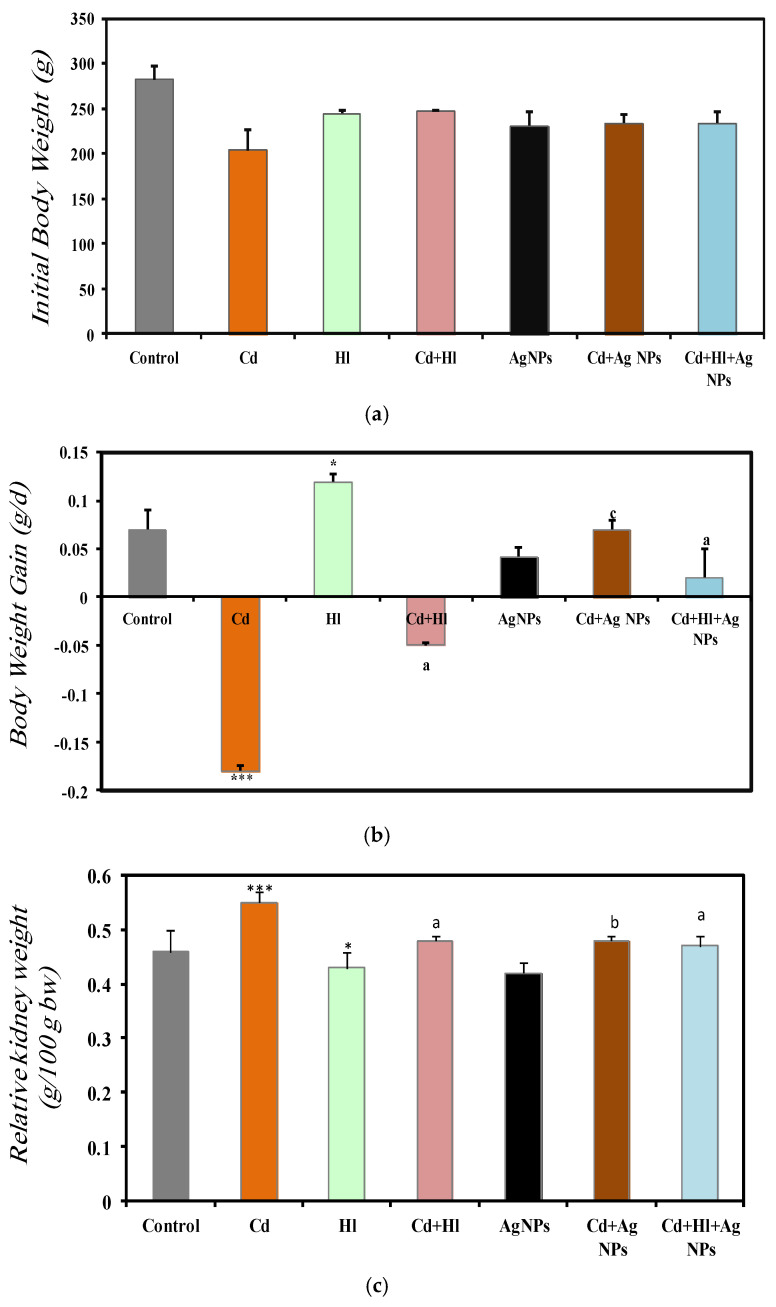
Comparison of baseline body weight (**a**), body weight gain (**b**), and relative kidney weight (**c**) in the seven rat groups. Group 1 (Control): no exposure/treatment. Group 2 (Cd): addition of CdCl_2_ (50 mg/kg body weight/day) in drinking water for 35 days. Group 3 (*Hl*): *H. lippii* extract (100 mg/kg body weight/day, by gavage, for 35 days). Group 4 (Cd + *Hl*): CdCl_2_ exposure (like in Group 2) followed by *H. lippii* extract (100 mg/kg body weight/day by gavage) for 15 days. Group 5 (Ag): Ag NPs (100 μg/kg body weight/day, intraperitoneal injection) for 35 days. Group 6 (Cd + Ag): CdCl_2_ exposure (like in Group 2) followed by Ag NPs (0.1 mg/kg body weight/day by intraperitoneal injection) for 15 days. Group 7 (Cd+ *Hl* +Ag): CdCl_2_ exposure (like in Group 2) followed by *H. lippii* extract (100 mg/kg body weight/day by gavage) and Ag NPs (0.1 mg/kg body weight/day by intraperitoneal injection) for 15 days. * *p* < 0.05, *** *p* < 0.001 vs. Group 1; a *p* < 0.05, b *p* < 0.01, c *p* < 0.001: vs. Group 2.

**Figure 4 pharmaceuticals-17-00982-f004:**
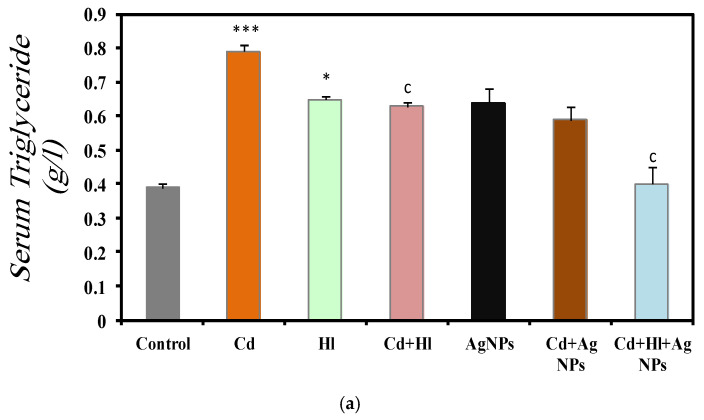
Comparison of different serum markers in the seven rat groups. Group 1 (Control): no treatment/exposure. Group 2 (Cd): exposure to CdCl_2_ (50 mg/kg body weight/day) in drinking water for 35 days. Group 3 (*Hl*): *H. lippii* extract (100 mg/kg body weight/day) by gavage for 35 days. Group 4 (Cd + *Hl*): CdCl_2_ exposure (like in Group 2) followed by *H. lippii* extract (100 mg/kg body weight/day by gavage) for 15 days. Group 5 (Ag): Ag NPs (100 μg/kg body weight/day, intraperitoneal injection) for 35 days. Group 6 (Cd + Ag): CdCl_2_ exposure (like in Group 2) followed by Ag NPs (0.1 mg/kg body weight/day by intraperitoneal injection) for 15 days. Group 7 (Cd + *Hl* + Ag): CdCl_2_ exposure (like in Group 2) followed by *H. lippii* extract (100 mg/kg body weight/day by gavage) and Ag NPs (0.1 mg/kg body weight/day by intraperitoneal injection) for 15 days. (**a**) serum triglycerides, (**b**) serum cholesterol, (**c**) serum glucose, (**d**) serum albumin. * *p* < 0,05, ** *p* < 0.01, *** *p* < 0.001 vs. Group 1; a *p* < 0.05, c *p* < 0.001: vs. Group 2.

**Figure 5 pharmaceuticals-17-00982-f005:**
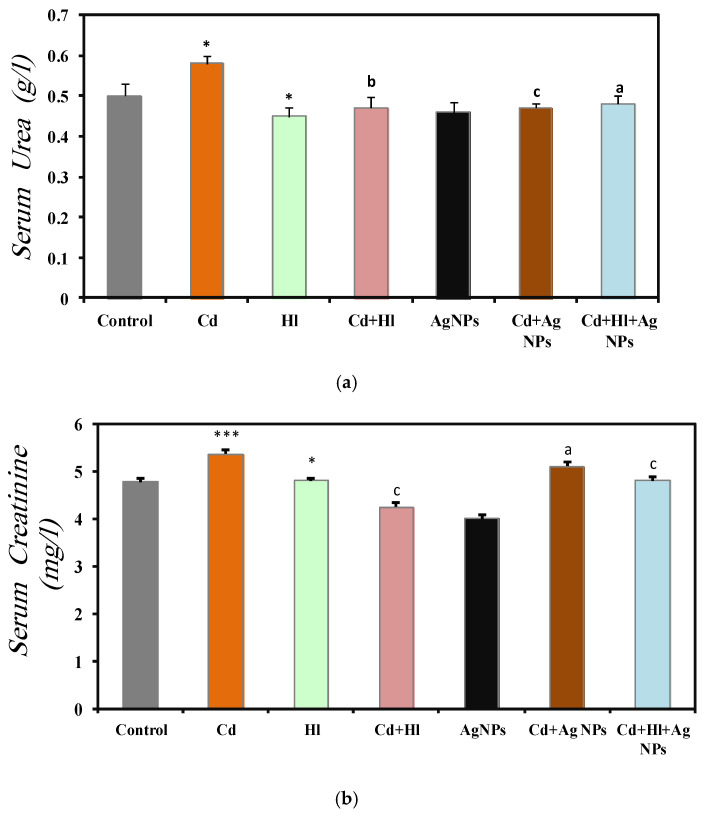
Renal function marker levels in the seven rat groups. Group 1 (Control): no treatment/exposure. Group 2 (Cd): addition of CdCl_2_ (50 mg/kg body weight/day) in drinking water for 35 days. Group 3 (Hl): *H. lippii* extract (100 mg/kg body weight/day), by gavage for 35 days. Group 4 (Cd + *Hl*): CdCl_2_ exposure (like in Group 2) followed by *H. lippii* extract (100 mg/kg body weight/day by gavage) for 15 days. Group 5 (Ag): Ag NPs (100 μg/kg body weight/day, intraperitoneal injection) for 35 days. Group 6 (Cd + Ag)/: CdCl_2_ exposure (like in Group 2) followed by Ag NPs (0.1 mg/kg, body weight/day by intraperitoneal injection) for 15 days. Group 7 (Cd + Hl + Ag): CdCl_2_ exposure (like in Group 2) followed by *H. lippii* extract (100 mg/kg, body weight/day by gavage) and Ag NPs (0.1 mg/kg, body weight/day by intraperitoneal injection) for 15 days. (**a**) serum urea, (**b**) serum creatinine, (**c**) serum uric acid, (**d**) serum blood ration urea/creatinine. * *p* < 0.05, *** *p* < 0.001 vs. Group 1; a *p* < 0.05, b *p* < 0.01, c *p* < 0.001: vs. Group 2.

**Figure 6 pharmaceuticals-17-00982-f006:**
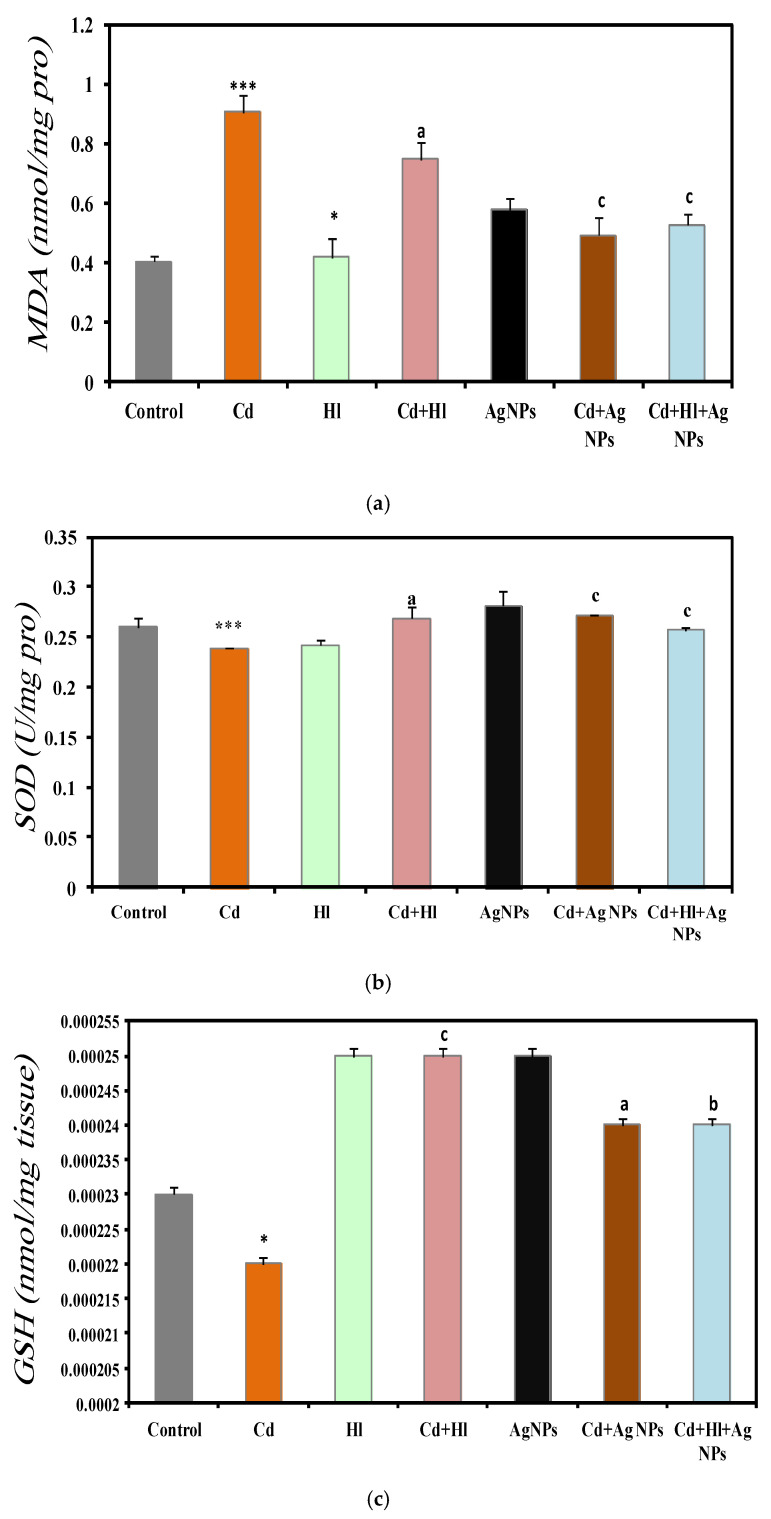
Oxidative stress markers in the seven rat groups. Group 1 (Control): no treatment/exposure. Group 2 (Cd): addition of CdCl_2_ (50 mg/kg body weight/day) in drinking water for 35 days. Group 3 (*Hl*): *H. lippii* extract (100 mg/kg body weight/day, by gavage for 35 days). Group 4 (Cd + *Hl*): CdCl_2_ exposure (like in Group II) followed by *H. lippii* extract (100 mg/kg body weight/day by gavage) for 15 days. Group 5 (Ag): Ag NPs (100 μg/kg body weight/day, by intraperitoneal injection) for 35 days. Group 6 (Cd + Ag): CdCl_2_ exposure (like in Group II) followed by Ag NPs (0.1 mg/kg body weight/day by intraperitoneal injection) for 15 days. Group 7 (Cd + Hl + Ag): CdCl_2_ exposure (like in Group II) followed by *H. lippii* extract (100 mg/kg body weight/day by gavage) and Ag NPs (0.1 mg/kg body weight/day by intraperitoneal injection) for 15 days. (**a**) MDA, (**b**) SOD, (**c**) GSH, (**d**) Catalase. * *p* <0.05, *** *p* < 0.001 vs. Group 1; a *p* < 0.05, b *p* < 0.01, c *p* < 0.001: vs. Group 2.

**Figure 7 pharmaceuticals-17-00982-f007:**
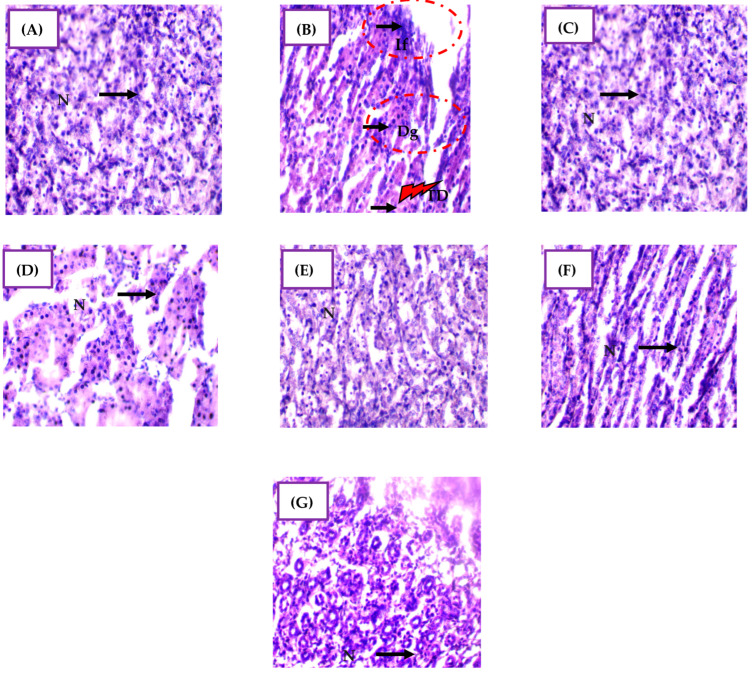
Micrographs of representative kidney tissue sections from rats in the different experimental groups, showing the effect of CdCl_2_ exposure and the mitigating effect of *H. lippii* and Ag NPs. (**A**) Group 1 (control); (**B**) Group 2 (CdCl_2_ exposure); (**C**) Group 3 *(H. lippii* extract alone); (**D**) Group 4 (*H. lippii* after CdCl_2_ exposure); (**E**) Group 5 (Ag NPs); (**F**) Group 6 (Ag NPs after CdCl_2_ exposure); (**G**) Group 7 (*H. lippii* and Ag NPs after CdCl_2_ exposure). N, normal cell; IF, inflammatory cells; DG, destroyed glomeruli. TD, Tubular dilatation; 40×.

**Table 1 pharmaceuticals-17-00982-t001:** Results of the phytochemical screening of the *H. lippii* aqueous extract.

Phytochemical Compounds		*H. lippii* (Aqueous Extract)
Polyphenols		(+)
Alkaloids	Mayer	(−)
	Wagner	(+)
Tannins	Catechin	(+)
	Gallic	(+)
Flavonoids		(+)
Saponins		(+)
Anthocyanins		(+)
Leucoanthocyanins		(+)
Cardiac glycosides		(+)
Steroids and terpenoids		(+)
Mucilages		(+)

(−) Absence. (+) Presence.

**Table 2 pharmaceuticals-17-00982-t002:** Retention time and concentration of phenolic compounds identified in *H. lippii* aqueous extract.

Phenolic Compound	Retention Time (Min)	Equation	Concentration (µg/g Extract)
Caffeic acid	16.27	y = 42,239x	444.81
p-Coumaric acid	23.81	y = 27,977	663.77
Gallic acid	5.29	y = 54,681x	9495.11
Vanillic acid	15.53	y = 20,674	ND
Chlorogenic acid	13.39	y = 21,665x	7107.24
Naringin	34.78	y = 19,379x	738.19
Rutin	28.37	y = 1649x	ND
Quercetin	45.04	y = 2,142,281x	1118.64
Vanillin	21.46	y = 9286x	ND

y: HPLC peak area. x: concentration (μg/mL), ND: not detected.

**Table 3 pharmaceuticals-17-00982-t003:** The sub-acute toxicity test of the dose of *H. lippii* aqueous extract and of the Ag NPs in Wistar albino rats. The dose was given in the time zero without exposing the Wistar rats to CdCl_2_, and the observation was made at different time intervales at 3 h, 24 h, 7 days, and 14 days.

Parameters	Dose of *H.lippii*				Dose of Ag NPs		
	Control Group (0 mg/kg)	Low Dose Group (100 mg/kg)	Medium Dose Group (1000 mg/kg)	High Dose Group (4000 mg/kg)	Control Group (0 mg/kg	Low Dose Group (2 mg/kg)	Medium Dose Group (10 mg/kg)
Body weight changes	N	N	N	N	N	N	N
Death	0	0	0	0	0	0	0
Movement	N	N	N	N	N	N	N
Diarrhea	N	N	N	N	N	N	N
Eyes	N	N	N	N	N	N	N

N: Normal.

## Data Availability

The original contributions presented in the study are included in the article/Appendix A, further inquiries can be directed to the corresponding author.

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
