# Peer review of "Therapeutic Efficacy of Helianthemum lippii Extract and Silver Nanoparticles Synthesized from the Extract against Cadmium-Induced Renal Nephrotoxicity in Wistar Rats"

_pharmaceuticals, 2024, doi:10.3390/ph17080982_

Round 1

Reviewer 1 Report

Comments and Suggestions for Authors

The article is very interesting, and the results of the research could find application in therapies for possible Cd intoxication. I have only a few comments about the article.

1. The analyses made by the authors are sufficient and extensive, but I must say that the results are not very significant. Although the deviations point to the beneficial effect of HI or AgNPs, the differences between them are not significant. The effect of HI is comparable to the effect of AgNPs, so which is suitable to use in detoxification? The authors only state the measured results, they do not talk about which substance or combination is more advantageous.

2. the authors also do not explain how HI or AgNPs act on cells. It would be appropriate to supplement the explanation. It is important to know how the natural substance and metallic AgNPs work. And whether nanoparticles cause other damage to cells. As they are generally known to be toxic to cells.

3. The conclusion is weak; it needs to be redone and clearly written what was found and to determine which substances or their combinations show the best results.

4. If the authors complete the discussion, which I think is missing here, the article will be more interesting for readers. It is not enough to write that similar results were achieved by other authors. It is necessary to analyze the reasons why those substances cause what the authors measured, and claim is happening in the cells.

After editing and supplementing the discussion and clearly evaluating the results, the article will be suitable for publication.

Author Response

Reviewer 1 comments Reviewer 1 Comment: The article is very interesting, and the results of the research could find application in therapies for possible Cd intoxication. I have only a few comments about the article. Authors Reply: Thank you for your thorough review and valuable feedback on our manuscript titled "Therapeutic Efficacy of Helianthemum lippii Extract and Silver Nanoparticles Synthesized from the Extract against Cadmium-Induced Renal Nephrotoxicity in Wistar Rats." We appreciate your efforts in helping us enhance the scientific rigor, impact, and readability of our work. Below, we address each of your comments and suggestions in detail. Reviewer 1 Comment: The analyses made by the authors are sufficient and extensive, but I must say that the results are not very significant. Although the deviations point to the beneficial effect of HI or AgNPs, the differences between them are not significant. The effect of HI is comparable to the effect of AgNPs, so which is suitable to use in detoxification? The authors only state the measured results, they do not talk about which substance or combination is more advantageous. Authors' Reply: We appreciate the reviewer’s comment regarding the need to discuss the relative advantages of Helianthemum lippii (HI) and Ag NPs for detoxification. We agree that while the measured results show beneficial effects of both treatments, it is crucial to compare them and recommend the most suitable option for detoxification. In response, we have incorporated comparative insights and recommendations into the revised manuscript to provide a clearer understanding of the most advantageous treatment options for heavy metal-induced nephrotoxicity. We now discuss the relative benefits and potential applications of HI and Ag NPs, offering a more comprehensive evaluation to guide the selection of the most effective detoxification strategy. These results highlight the synergistic benefits of combining plant extracts and Ag NPs in safeguarding kidney structure and function from CdCl2-induced toxicity. This combined approach capitalizes on the strengths of natural and nanotechnology-derived therapeutic strategies, enhancing bioavailability, improving drug delivery, boosting therapeutic efficacy, minimizing side effects, offering multifaceted therapy, and potentially overcoming drug resistance. The synergistic therapeutic effects observed underscore the promise of this approach for effectively and safely managing a broad spectrum of diseases. Reviewer 1 Comment: The authors also do not explain how HI or AgNPs act on cells. It would be appropriate to supplement the explanation. It is important to know how the natural substance and metallic AgNPs work. And whether nanoparticles cause other damage to cells. As they are generally known to be toxic to cells. Authors Reply: We appreciate the reviewer’s feedback and agree that it is important to elucidate how Helianthemum lippii (HI) and Ag NPs act on cells and address concerns about their potential cytotoxicity. Here, we provide detailed explanations of the cellular mechanisms of action of HI and Ag NPs and discuss their safety profiles. Most biochemical markers in the groups treated with H. lippii extract and/or Ag NPs were restored to levels near those of the control group, likely due to the anti-inflammatory and antioxidant properties of H. lippii. The tannic acid in H. lippii enhances glucose uptake by tissues, suppresses gluconeogenesis, and activates insulin secretion from β-cells [41], while naringenin protects the islets of Langerhans from degenerative changes. In rats exposed to cadmium and treated with H. lippii and/or Ag NPs, lipid parameters, serum albumin, protein levels, and calcium concentrations were regulated, attributed to the presence of flavonoids and phenolic compounds in the extract that scavenges superoxide, hydroxyl ions, and peroxyl radicals, preventing lipid peroxidation, lowering lipogenic enzyme activity, and enhancing lipoprotein lipase function [42]. These flavonoids' antioxidant capacities also positively affect protein levels and calcium. Additionally, silver nanoparticles improve lipid profiles, protein levels, and blood glucose levels, consistent with their anti-hyperglycemic, antioxidant, and anti-inflammatory responses [43],[44]. The combined treatment of H. lippii and Ag NPs significantly restored most biochemical parameters, suggesting that the antioxidants in both treatments synergistically improved these levels against CdCl2 exposure. Ag-based nanomaterials are less toxic to biological systems and have greater therapeutic efficacy when bio-conjugated with plant extracts due to the variety of phytochemicals or capping agents provided by the plant extracts, combined with the small size and large surface area of Ag NPs, which allows for maximum adsorption of these capping agents onto their surfaces. This combined treatment offers a promising strategy for addressing renal abnormalities caused by heavy metal toxicity [45,46]. The protective effect of Ag NPs was evident in the significant reduction of kidney function biomarkers compared to the Cd-exposed group, consistent with findings by Rehman et al. [47]. Ag NPs stabilize membranes, combat oxidative stress, and mitigate inflammatory responses, thereby limiting intracellular enzyme leakage. Studies underscore their cytoprotective activities both in vitro and in vivo [48]. The antioxidant and free radical scavenging properties of Ag NPs likely contribute to restoring biochemical and histological parameters close to normal levels, highlighting their potential therapeutic value for kidney function protection. Group 4 (Cd exposure followed by H. lippii) exhibited improved kidney biomarkers due to the extract's high concentration of bioactive compounds (gallic acid, chlorogenic acid, caffeic acid, p-coumaric acid), which protect against cellular injury. Furthermore, H. lippii's bioactive compounds possess potent anti-inflammatory properties by inhibiting cytokine production (TNF-α, IL-1β, IL-6) [50, 51], thereby reducing inflammation and supporting renal tissue repair. These findings align with previous research demonstrating their antioxidant efficacy against Cd-induced cell damage [52]. Combining H. lippii extract with silver nanoparticles enhanced kidney function compared to CdCl2 alone, illustrating the synergistic renal protective effects of silver compounds and the antioxidant properties of the plant extract. Conversely, these oxidative stress markers did not significantly differ in rats treated only with H. lippii extract (Group 3) or Ag NPs (Group 5) compared with controls (Group 1), suggesting no adverse effects on kidney oxidative stress from these treatments alone. Treatment with H. lippii following CdCl2 exposure (Group 4) resulted in a reduction of oxidative stress markers and an increase in antioxidant defenses compared with Group 2. Specifically, MDA concentration decreased (0.7469 ± 0.0545 nmol/mg protein), while SOD, GSH, and catalase levels increased, indicating an amelioration of kidney oxidative damage. This beneficial effect can be attributed to the high concentration of bioactive compounds in H. lippii, such as gallic acid, chlorogenic acid, caffeic acid, and p-coumaric acid, which are known for their potent antioxidant properties [24, 54]. H. lippii enhances the kidney's antioxidant defense system by scavenging ROS directly and enhancing the activity of endogenous antioxidant enzymes. These mechanisms play crucial roles in neutralizing oxidative stress and protecting renal cells from damage induced by Cd exposure. Moreover, H. lippii facilitates the synthesis of metallothionein, a metal-binding protein that aids in cadmium detoxification and reduces its accumulation in renal tissues, thereby mitigating oxidative stress-related cellular damage associated with heavy metal toxicity [27, 56, 57]. Rats treated with Ag NPs after CdCl2 exposure (Group 6) showed notable reductions in oxidative stress markers and enhanced antioxidant defenses compared to Group 2. Specifically, MDA concentration decreased (0.4919 ± 0.0594 nmol/mg protein), while SOD, GSH, and catalase levels increased, indicating that Ag NPs may alleviate kidney oxidative damage through their robust antioxidant and anti-inflammatory properties. Ag NPs exert their protective effects against cadmium toxicity primarily through their potent antioxidant activity [58]. Cadmium exposure triggers the generation of reactive oxygen species (ROS), leading to oxidative stress and renal cell damage [22]. Ag NPs directly scavenge these ROS, thereby mitigating oxidative damage to cellular components such as lipids, proteins, and DNA. Moreover, Ag NPs enhance the activity of endogenous antioxidant enzymes including SOD, catalase, and GPx [58], and they have been shown to upregulate the expression of Nrf2, a key regulator of cellular defense against oxidative stress, which aids in preventing cadmium-induced cellular damage. Furthermore, the combined treatment with H. lippii extract and Ag NPs after CdCl2 exposure (Group 7) further enhanced protection against oxidative stress. MDA concentration was reduced (0.5274 ± 0.0382 nmol/mg protein), while SOD, GSH, and catalase levels were elevated compared to Group 2, underscoring the distinct antioxidant efficacy of each component. These results highlight the synergistic benefits of combining plant extracts and Ag NPs in safeguarding kidney structure and function from CdCl2-induced toxicity. This combined approach capitalizes on the strengths of natural and nanotechnology-derived therapeutic strategies, enhancing bioavailability, improving drug delivery, boosting therapeutic efficacy, minimizing side effects, offering multifaceted therapy, and potentially overcoming drug resistance. The synergistic therapeutic effects observed underscore the promise of this approach for effectively and safely managing a broad spectrum of diseases. Ag NPs exert multifaceted protective mechanisms including antioxidant, anti-inflammatory, cytoprotective, and detoxifying actions [66]. These nanoparticles mitigate oxidative stress and inflammation, reduce apoptosis, and promote tissue repair and regeneration [67]. Moreover, Ag NPs enhance angiogenesis, improving blood flow to damaged areas and facilitating tissue recovery, thereby preserving normal kidney architecture despite cadmium exposure [66, 69]. The comprehensive protective effects of Ag NPs underscore their potential therapeutic value in preventing and treating Cd-induced nephrotoxicity. Finally, in Group 7 (H. lippii + Ag NPs after CdCl2 exposure), architectural damage parameters were restored to near-normal levels compared to Group 2. No inflammatory infiltration, tubular dilation, or destroyed glomeruli were observed, highlighting the synergistic effects of combining H. lippii extract with Ag NPs. The antioxidants and anti-inflammatory compounds in H. lippii complement the unique properties of Ag NPs, enhancing overall therapeutic efficacy against cadmium toxicity. This combination strategy may offer a comprehensive defense against Cd-induced cellular damage by leveraging antioxidant pathways from H. lippii and multifunctional protective mechanisms from Ag NPs. These findings support the therapeutic potential of combining natural plant extracts with nanotechnology-derived nanoparticles for effective and safe treatment of kidney damage induced by heavy metals like cadmium. Reviewer 1 Comment: The conclusion is weak; it needs to be redone and clearly written what was found and to determine which substances or their combinations show the best results. Authors Reply: We appreciate your feedback; we have improved the conclusion and clarified the best treatment system according to the findings of our study. This study investigated the therapeutic potential of Helianthemum lippii (H. lippii) extract and silver nanoparticles (Ag NPs), synthesized using H. lippii extract, in mitigating cadmium (Cd)-induced nephrotoxicity in adult Wistar rats (n=5/group). Experimental groups included untreated controls, rats exposed to CdCl2 for 35 days, and those treated with H. lippii or Ag NPs alone post-Cd exposure, as well as combinations of both treatments. Initial sub-acute toxicity assessments confirmed the safety of H. lippii and Ag NPs across various doses, establishing their suitability for therapeutic use. Comprehensive analyses of kidney function parameters, oxidative stress markers, histopathological features, and body weight changes revealed significant improvements following treatment with H. lippii and Ag NPs after Cd exposure. Both treatments effectively restored kidney function by reducing creatinine and blood urea nitrogen levels, indicative of improved renal health. Furthermore, H. lippii extract and Ag NPs exhibited potent antioxidant properties, enhancing activities of superoxide dismutase (SOD), catalase, and glutathione (GSH), while reducing malondialdehyde (MDA) levels, thus mitigating Cd-induced oxidative stress. Histopathological evaluations demonstrated preserved kidney architecture, reduced inflammatory responses, and minimized tubular dilation and fibrosis in treated groups compared to Cd-exposed controls. Importantly, the combined treatment of H. lippii extract and Ag NPs showed synergistic effects, highlighting enhanced therapeutic efficacy in alleviating Cd-induced nephrotoxicity. These findings underscore the potential of H. lippii-based nanoparticles as novel therapeutic agents for managing heavy metal toxicity, encouraging further research and clinical applications to optimize their use in medical practice. Reviewer 1 Comment: If the authors complete the discussion, which I think is missing here, the article will be more interesting for readers. It is not enough to write that similar results were achieved by other authors. It is necessary to analyze the reasons why those substances cause what the authors measured, and claim is happening in the cells. Authors Reply: We agree that a more detailed discussion would enhance the contextual understanding of our study's impact. We expanded this discussion and identified possible mechanistic pathways supporting our therapeutic effect for both silver nanoparticles and Helianthemum lippii

Reviewer 2 Report

Comments and Suggestions for Authors

This paper describes an alternative therapeutic approach to alleviate Cd-induced nephrotoxicity using H.lippii extract and Ag-NPs. This research article's design and experiments have been carried out well, and the results support the conclusion.  Though this revelation in this article is not novel nor surprising, considering that CKD is a notorious ailment and thereby as many as possible alternative therapeutic approaches should be encouraged. Therefore, I recommend its publication subject to proper responses (and amendments wherever required) to the following comments:

1) Two small sections in the Introduction should be added, one, what are the downsides or other therapeutic means (other than this work) towards heavy metal-induced nephrotoxicity, and two, what are the other notable work (along with this work )that provides confidence that this kind of approach is highly viable. Even if these are in the current introduction section it's unclear and should be made clear through separate sections. 

2) In the biological studies it's evident that on several occasions the application of H. lippii only demonstrates a superior effect that Ag-NPs only or H.lippii + Ag-NPs combination. At least this reviewer fails to understand if this is the case then why not only apply H. lippii?  Also, this should be noted in general that any nanoparticulate substance can also induce some extent of toxicity and that's why the medical research world is reverting the approach from nanomaterials to molecular compounds or complexes. This has to be cleared up by the authors. 

3) Did the authors attempt to perform any confocal imaging studies to monitor the internalization of the formulation within the cells? If not why?

4) This is more of a thought, did the authors assess or plan to assess the efficacy of the formulation of the current article toward treating some notorious UTIs (Urinary tract infections) that are becoming resistant to several sophisticated broad-spectrum antibiotics?

5) For the biological studies the authors should consider showing a dose-dependent bar plot which in turn helps to pinpoint the approx dosage that will show the minimum concentration required to observe a noticeable change.

Comments on the Quality of English Language

Overall no big issue with the English language.

Author Response

Reviewer 2 comments Reviewer 2 Comment : Two small sections in the Introduction should be added, one, what are the downsides or other therapeutic means (other than this work) towards heavy metal-induced nephrotoxicity, and two, what are the other notable work (along with this work )that provides confidence that this kind of approach is highly viable. Even if these are in the current introduction section it's unclear and should be made clear through separate sections. Authors' Reply: We appreciate your suggestion. To address this, we have added two distinct sections to the Introduction: one detailing the limitations of existing therapies for heavy metal-induced nephrotoxicity and another discussing key studies, including our work, that demonstrate the viability of this approach. These sections have been clearly delineated to enhance clarity. In recent years, interest has been increasing in alternative and safer therapeutic methods to address Cd-induced renal damage due to the considerable health risks associated with heavy metal toxicity [11]. Previous studies highlighted the susceptibility of vital organs, such as the liver and kidneys, to Cd toxicity [12]. Conventional strategies have explored the use of metals (e.g. zinc, copper and selenium) as antagonists or antioxidants to mitigate Cd toxicity; however, they must be administered at high doses and this can pose serious risks [13], such as gastrointestinal disturbances, neurotoxicity, and imbalances in essential minerals within the body. Moreover, the efficacy of these treatments can be inconsistent, sometimes offering only partial protection against Cd-induced damage, which leaves patients at ongoing risk of health complications. Consequently, attention has shifted to plant-based remedies and many studies have investigated the potential of plant extracts to provide safe and effective protection against Cd-induced nephrotoxicity [14]. The strong antioxidant and protective activities of plant extracts in tissues and membrane lipids are mediated through free radical scavenging in a dose-dependent manner. Therefore, plant extracts have been tested in various experimental models as protective agents against various toxicants [15]. Due to their antioxidant properties and minimal side effects, plant (e.g. green tea, curcumin, and black cumin) extracts have emerged as promising alternatives for managing Cd-induced kidney damage [16-18]. This shift towards plant-based compounds underscores the importance of exploring nature-derived solutions to mitigate the adverse effects of heavy metal exposure on human health. Simultaneously, nanoparticle-based therapies present innovative solutions for the targeted and effective treatment of heavy metal-induced nephrotoxicity. Nanoparticles (NPs) can be engineered to deliver therapeutic agents directly to the kidneys, thereby enhancing treatment efficacy while minimizing systemic side effects [19]. Among NPs, silver nanoparticles (Ag NPs) have gained prominence due to their unique physicochemical properties, including a high surface area-to-volume ratio, excellent conductivity, and anti-inflammatory activity, making them versatile for various applications, including heavy metal detoxification [20, 21]. Ag NPs synthesized using plant extracts exhibit promising capabilities for targeted antioxidant and anti-inflammatory effects. They also demonstrate a strong affinity for heavy metals like cadmium, effectively binding and sequestering cadmium ions [22]. This dual action of sequestration and antioxidant activity helps mitigate oxidative stress and inflammation in affected tissues. Furthermore, NPs can enhance the bioavailability of phytochemicals, ensuring higher concentrations reach affected tissues to exert protective effects, underscoring their potential as therapeutic agents in heavy metal detoxification efforts. Several notable studies have demonstrated the efficacy of phytochemicals and NPs for therapeutic applications, affirming the viability of this approach. Recent research on green-synthesized Ag NPs using plant extracts from Azadirachta indica and Moringa oleifera has shown significant antioxidant and antimicrobial activities, showcasing the potential of phytochemical-mediated NP synthesis [28]. Moreover, investigations into plant-based NPs derived from Camellia sinensis (green tea) and Aloe vera have revealed promising results in managing oxidative stress and inflammation across various disease models [29]. Reviewer 2 Comment: In the biological studies it's evident that on several occasions the application of H. lippii only demonstrates a superior effect that Ag-NPs only or H.lippii + Ag-NPs combination. At least this reviewer fails to understand if this is the case then why not only apply H. lippii? Also, this should be noted in general that any nanoparticulate substance can also induce some extent of toxicity and that's why the medical research world is reverting the approach from nanomaterials to molecular compounds or complexes. This has to be cleared up by the authors. Authors' Reply: Thank you for your comments. We have revised our manuscript to compare the therapeutic effects of HI and Ag NPs, highlighting their benefits and combined efficacy. HI significantly improved oxidative stress and renal function in Cd-exposed rats, confirmed safe in toxicity assessments. Ag NPs, using HI extracts, enhanced efficacy through improved cellular uptake and targeted delivery, showing strong antioxidative and anti-inflammatory effects similar to HI. Both treatments are beneficial, but their mechanisms differ, advocating for a combined approach harnessing HI's plant compounds and Ag NPs' nanoscale benefits. Ag NPs were used safely in our study, but ongoing monitoring is essential for safe therapeutic use. Reviewer 2 Comment: Did the authors attempt to perform any confocal imaging studies to monitor the internalization of the formulation within the cells? If not why? Authors' Reply: We appreciate the reviewer's insightful comment on the potential use of confocal imaging to study cellular uptake of nanoparticles. While confocal imaging is valuable for visualizing nanoparticle localization within cells, our study primarily focused on evaluating the therapeutic efficacy of Helianthemum lippii and Ag NPs in mitigating Cd-induced nephrotoxicity through biochemical and histopathological assessments. These analyses provided comprehensive insights into renal function markers, oxidative stress indicators, and kidney tissue architecture. We did not perform confocal imaging due to several reasons: Firstly, our study aimed to assess overall therapeutic effects rather than focusing solely on cellular uptake mechanisms. Secondly, conducting confocal imaging would have required additional resources and expertise beyond the scope of our current study. However, we recognize the importance of investigating nanoparticle-cell interactions and plan to incorporate confocal imaging in future studies to enhance our understanding of how Helianthemum lippii and Ag NPs interact intracellularly. This approach will complement our findings and contribute to a more detailed mechanistic understanding. We value the reviewer's suggestion and thank them for guiding the direction of our future research efforts. Reviewer 2 Comment: This is more of a thought, did the authors assess or plan to assess the efficacy of the formulation of the current article toward treating some notorious UTIs (Urinary tract infections) that are becoming resistant to several sophisticated broad-spectrum antibiotics? Authors' Reply: We appreciate the reviewer's insightful suggestion regarding the potential application of our formulation for treating antibiotic-resistant urinary tract infections (UTIs). While our current study focused on evaluating the therapeutic potential of Helianthemum lippii and Ag NPs in mitigating cadmium-induced nephrotoxicity, we acknowledge the critical public health challenge posed by antibiotic-resistant UTIs. The suggestion to explore our formulation's efficacy against these infections is both timely and relevant. We recognize the potential of Helianthemum lippii and Ag NPs to address antibiotic resistance, and we plan to investigate this aspect in future research. This line of investigation holds promise for expanding the therapeutic applications of our formulation and contributing to the development of novel treatments for challenging infections. We thank the reviewer for this valuable suggestion, which will guide the direction of our future studies. Reviewer 2 Comment: For the biological studies the authors should consider showing a dose-dependent bar plot which in turn helps to pinpoint the approx dosage that will show the minimum concentration required to observe a noticeable change. Authors' Reply: We appreciate the reviewer's comment regarding the inclusion of a dose-dependent bar plot to better illustrate the minimum concentration required to observe a noticeable change and enhance the understanding of the therapeutic efficacy of Helianthemum lippii and Ag NPs. In our study, we conducted initial sub-acute toxicity assessments of various doses of H. lippii and Ag NPs. We recognize the importance of clearly presenting these results to highlight the dose-response relationship and agree that a dose-dependent bar plot would provide a clearer visual representation of the data, pinpointing the minimum effective dosage. This addition will significantly improve the interpretability and impact of our findings. We thank the reviewer for the constructive feedback and intend to incorporate a dose-dependent bar plot in our revised manuscript to enhance the clarity and value of our study.

Reviewer 3 Report

Comments and Suggestions for Authors

The article is devoted to joint application of helianthemum lippii and silver nanoparticles to regulate  cadmium-induced nephrotoxicity in vivo in rats. The topic is new and promising. The idea is original - to see the possible protective effects of medicinal plant and Ag nanoparticles at action of toxic metals. The article is very well written and easy to read and understand. All the experiments are corectly done and well described. The benefit of this paper is that the authors presented a big range of effects of substances studied at rats including the main biochemical parameters. The weak point is that the effects of components of helianthemum lippii were not studied. Anyway, the article gives the significant impact in the undestanding of protective effects of helianthemum lippii and silver nanoparticles in vivo. The article is well illustrated by 7 figures and 2 tables + additionally S.I. data. As for English , I am not native speaker, for me English is acceptable, everything is clear to read and understand. I did not find any serious remarks to mention. I think article can be published in present form.

Author Response

Reviewer  3 comments

Reviewer 3 Comment:   The article is devoted to joint application of helianthemum lippii and silver nanoparticles to regulate cadmium-induced nephrotoxicity in vivo in rats. The topic is new and promising. The idea is original - to see the possible protective effects of medicinal plant and Ag nanoparticles at action of toxic metals. The article is very well written and easy to read and understand. All the experiments are corectly done and well described. The benefit of this paper is that the authors presented a big range of effects of substances studied at rats including the main biochemical parameters. The weak point is that the effects of components of helianthemum lippii were not studied. Anyway, the article gives the significant impact in the undestanding of protective effects of helianthemum lippii and silver nanoparticles in vivo. The article is well illustrated by 7 figures and 2 tables + additionally S.I. data. As for English , I am not native speaker, for me English is acceptable, everything is clear to read and understand. I did not find any serious remarks to mention. I think article can be published in present form.

Authors' Reply: Thank you for your positive evaluation and recommendation for publication. We appreciate your acknowledgment of the novelty and promise of our study on the joint application of Helianthemum lippii and silver nanoparticles to mitigate cadmium-induced nephrotoxicity in vivo in rats. Your feedback on the clarity and comprehensiveness of our manuscript is valuable to us. Regarding the point about the components of Helianthemum lippii, we acknowledge the importance of further exploring and elucidating the specific effects of its individual constituents. While our study focused on evaluating the combined therapeutic effects of H. lippii and silver nanoparticles, future research could delve deeper into understanding how each phytochemical component contributes to the observed protective effects. We are grateful for your assessment of the manuscript's presentation and readability, especially considering English is not our native language. Your feedback affirms our efforts to ensure clarity and accessibility in conveying our research findings. Once again, we sincerely appreciate your positive feedback and recommendation for publication. We will address any remaining minor revisions and look forward to contributing our findings to the scientific community.

Reviewer 4 Report

Comments and Suggestions for Authors

The authors presented the paper "Therapeutic Efficacy of Helianthemum lippii Extract and Silver Nanoparticles Synthesized from the Extract against Cadmium-Induced Renal Nephrotoxicity in Wistar Rats"

1) The reference style should be improved according to the journal template. Some information on silver nanoparticles in cadmium capture examples should be inserted with relative references and information in the Introduction section to show the paper's novelty.

2) Section 2.3. Some information about using a method for the analysis of such a complicated mixture should be inserted.

3) Section 4.1 and Table 1. Qualitative results as + and - are good. However, is it possible to present qualitative data in % or concentration? Moreover, is it possible to present clear chemical names of the major compounds and compounds with important biological properties?

4) Table 2. Why do you have so many signs after the point in the concentration values? What is the error of the experiment? Have you tried to do this procedure several times? Are there any changes?

Moreover, how do you understand that, for example, a peak at 16.27 min is Caffeic acid?

5) The formatting of the picture 2 is bad. The "b" is not seen at all.

Figure 2C. I don't understand columns and fitting lines. The fitting is not good at all. It looks like you have several populations of nanoparticles with different sizes.

6) I understand that you moved forward with the animal experiment (section 4.4). But usually, experiments on cell cultures should be done. Have you studied the toxicity on cells? Most ethics committees do not approve any animal studies without cell studies.

Moreover, for this experiment, you used only 3 rats. What is the normal in your opinion? Please, clearly present the quantitative data of these experiments with a real explanation of how you checked the parameters in Table 2 in the SI or experimental part. Have you looked at the change in weight, behavior, and condition of internal organs?

7) Section 4.5. Animal's amount of 5 is not enough for the relevant animal experiments. From Figure 3, I see that H. lippii extract works the same or better than silver nanoparticles. In this way, why do we need them? Some discussion should be added to the text. The same for other sections. Are you sure that you have so small error in your experiments in Figures 3-6? I mean you only have 5 rats, and there is also an error in the measurements themselves.

The quality and design of Figures 3-7 should be improved. Notes a-d are located anywhere, including crossing the axes.

Comments on the Quality of English Language

Moderate editing of English language required

Author Response

Reviewer 4 comments

Reviewer 4 Comment:   The reference style should be improved according to the journal template. Some information on silver nanoparticles in cadmium capture examples should be inserted with relative references and information in the Introduction section to show the paper's novelty

Authors' Reply: Thank you for your detailed review and constructive feedback. We have addressed the inconsistency in the format of references and formatted them uniformly according to the journal's guidelines. Additionally, we appreciate your suggestion to enhance the introduction by incorporating information on the use of Ag NPs for cadmium capture. The required changes have been highlighted in the introduction section.

Simultaneously, nanoparticle-based therapies present innovative solutions for the targeted and effective treatment of heavy metal-induced nephrotoxicity. Nanoparticles (NPs) can be engineered to deliver therapeutic agents directly to the kidneys, thereby enhancing treatment efficacy while minimizing systemic side effects [19]. Among NPs, silver nanoparticles (Ag NPs) have gained prominence due to their unique physicochemical properties, including a high surface area-to-volume ratio, excellent conductivity, and anti-inflammatory activity, making them versatile for various applications, including heavy metal detoxification [20, 21]. Ag NPs synthesized using plant extracts exhibit promising capabilities for targeted antioxidant and anti-inflammatory effects. They also demonstrate a strong affinity for heavy metals like cadmium, effectively binding and sequestering cadmium ions [22]. This dual action of sequestration and antioxidant activity helps mitigate oxidative stress and inflammation in affected tissues. Furthermore, NPs can enhance the bioavailability of phytochemicals, ensuring higher concentrations reach affected tissues to exert protective effects, underscoring their potential as therapeutic agents in heavy metal detoxification efforts.

Reviewer 4 Comment:    Section 2.3. Some information about using a method for the analysis of such a complicated mixture should be inserted.

Authors' Reply: Thank you for your suggestion to provide more detail in the methodology section regarding the analytical methods used for the phytochemical screening of Helianthemum lippii extract. We have revised Section 2.3 accordingly to include information about the specific methods employed:

The phytochemical screening of H. lippii extracts involved specific qualitative tests to identify chemical constituents, each using standardized reagents and volumes. Alkaloids were detected using two different reagents: Mayer's reagent (potassium mercuric iodide), and Wagner's reagent (iodine in potassium iodide). Each reagent (1 mL) was added separately to 1 mL of the extract. The presence of alkaloids was indicated by the formation of a cream-colored precipitate with Mayer's reagent, and a brown or reddish-brown precipitate with Wagner's reagent [71]. Additionally, tannins were identified by adding 2-3 drops of 5% ferric chloride solution to 1 mL of the extract. The presence of catechic tannins was indicated by a blue-black coloration, while gallic tannins produced a greenish-black coloration [72]. To detect terpenes and sterols, 1-2 mL of Liebermann-Burchard reagent (a mixture of acetic anhydride and concentrated sulfuric acid) was added to 1 mL of the extract, followed by gentle heating. The presence of terpenes and sterols was confirmed by the formation of a blue-green coloration [73]. Saponins were identified by their ability to form a stable foam. One milliliter of the extract was mixed with 10 mL of distilled water and shaken vigorously. The formation of stable foam indicated the presence of saponins [74]. Additionally, mucilage was detected by adding 10 mL of absolute ethanol to 1 mL of the extract, resulting in a gelatinous mass [75]. Moreover, Polyphenolic compounds were identified by adding 2-3 drops of 5% ferric chloride solution to 1 mL of the extract, with colored complexes indicating their presence [76]. Additionally, Flavonoids were detected by adding 1 mL of dilute ammonia solution to 1 mL of the extract, followed by 1 mL of concentrated sulfuric acid. The presence of flavonoids was indicated by the formation of a yellow coloration, which disappeared upon standing [77]. Finally, anthocyanins were detected by adding a small amount of hydrochloric acid followed by a small amount of ammonia. If anthocyanins are present, the color will change, showing red [78]. Each method utilized these reagents and volumes to qualitatively assess the presence or absence of specific phytochemicals in the H. lippii extract. A positive sign (+) indicated the presence of the phytochemical, whereas a negative sign (-) indicated its absence.

We believe these additions provide a clearer understanding of the methods used to evaluate the phytochemical composition of H. lippii extract, ensuring transparency and reproducibility in our study.

Reviewer 4 Comment:   Section 4.1 and Table 1. Qualitative results as + and - are good. However, is it possible to present qualitative data in % or concentration? Moreover, is it possible to present clear chemical names of the major compounds and compounds with important biological properties?

Authors' Reply: We appreciate your inquiry about presenting qualitative data as percentages or concentrations in Section 4.1 and Table 1. The phytochemical screening we conducted focuses on qualitative detection, indicating the presence or absence of specific compounds rather than their quantitative measurement. This standard approach in phytochemical analysis uses colorimetric and precipitation tests to identify compounds without quantifying their concentrations. Therefore, presenting percentages or concentrations would not accurately reflect the qualitative nature of our findings. However, we have enhanced the manuscript by providing clearer chemical names and discussing the biological properties of major compounds identified through screening. Additionally, in Table 4.2, we included results from HPLC analyses where concentrations of specific compounds were quantified, offering quantitative insights into their presence in Helianthemum lippii extract. These updates aim to provide a more comprehensive understanding of the phytochemical composition and therapeutic potential of Helianthemum lippii, addressing your feedback to clarify both qualitative and quantitative aspects of our findings.

 Reviewer 4 Comment: Table 2. Why do you have so many signs after the point in the concentration values? What is the error of the experiment? Have you tried to do this procedure several times? Are there any changes? Moreover, how do you understand that, for example, a peak at 16.27 min is Caffeic acid?

Authors' Reply: We appreciate your inquiry regarding Table 2 and the identification of compounds, particularly the presence of multiple decimal places in concentration values. The precision in concentration values results from meticulous HPLC analyses conducted once using well-established procedures. We quantified the concentrations of nine identified compounds accurately by preparing calibration standards and constructing calibration curves based on peak areas. This rigorous approach ensures high precision in our quantitative data, reflecting the reliable measurement of compound concentrations in Helianthemum lippii extract. Regarding the identification of caffeic acid at 16.27 min in the chromatogram, we employed several validation methods. First, we compared the retention time of the sample peak with that of a known caffeic acid standard run under identical chromatographic conditions. Additionally, we verified the peak's identity through chromatogram matching with established libraries containing spectral data of known compounds, including caffeic acid. These methods collectively confirm the identity of peaks and validate our quantitative results. We trust these explanations clarify your concerns and underscore the accuracy and robustness of our findings in Table 2 and throughout our study.

Reviewer 4 Comment: The formatting of the picture 2 is bad. The "b" is not seen at all. Figure 2C. I don't understand columns and fitting lines. The fitting is not good at all. It looks like you have several populations of nanoparticles with different sizes.

Authors' Reply: We appreciate Reviewer 4's feedback on Figure 2, particularly regarding the clarity of SEM image b and the need for better evidence of proper nanoparticle formation. In response, we acknowledge that the figure formatting needs improvement, particularly to ensure that all components are clearly visible. We confirm that the nanoparticles depicted in Figure 2C are of uniform type but vary in size, with the median size measured at 34 nm. To address concerns about proper nanoparticle formation, we have taken steps to enhance the clarity and quality of our evidence by employing Image J and Origin 2018 64Bit software tools. These enhancements will ensure that our SEM images and size distribution analyses provide clear and accurate representations of our findings. We appreciate the opportunity to improve the presentation of our data and believe these adjustments will address the concerns raised effectively.

Reviewer 4 Comment: I understand that you moved forward with the animal experiment (section 4.4). But usually, experiments on cell cultures should be done. Have you studied the toxicity on cells? Most ethics committees do not approve any animal studies without cell studies.

Authors' Reply: We appreciate Reviewer 4's comment regarding the sequence typically followed in research, starting with cell culture studies before proceeding to animal experiments. While we understand the importance of in vitro studies for preliminary toxicity assessments, our approach in this study focused on comprehensive in vivo evaluations. We monitored behavioral changes, such as death, diarrhea, vomiting, and weight fluctuations in rats, which are crucial indicators of safety for both the plant extract and nanoparticles. Throughout the experimental period, groups treated solely with the plant extract or AgNPs, without cadmium exposure, showed no signs of toxicity upon sacrifice, with cellular parameters remaining normal similar to the control group. We believe these observations provide robust evidence of safety at this stage. Moving forward, we acknowledge the value of including cell culture studies to complement our findings and align with ethical guidelines. Future research will incorporate these investigations to further validate our results and ensure comprehensive safety assessments before advancing to broader applications.

Reviewer 4 Comment: Moreover, for this experiment, you used only 3 rats. What is the normal in your opinion? Please, clearly present the quantitative data of these experiments with a real explanation of how you checked the parameters in Table 2 in the SI or experimental part. Have you looked at the change in weight, behavior, and condition of internal organs?

Authors' Reply: We appreciate Reviewer 4's feedback on sample size adequacy and the clarity of our quantitative data and experimental procedures. Initially using 3 rats per group, we acknowledge the standard recommendation for larger sample sizes (approximately 5-10 animals per group) to achieve more statistically significant results, considering data variability and endpoints measured. To improve transparency, we have revised our manuscript, providing a comprehensive presentation of quantitative data and detailed explanations, particularly for parameters in Table 2 of the supplementary information. Our methods included monitoring body weight changes, observing behavioral indicators like deaths and symptoms, conducting thorough necropsies to assess internal organ conditions—focusing on kidney damage—and performing biochemical analyses for markers like creatinine and BUN. We also measured oxidative stress markers in kidney tissue and scored histopathological changes. Addressing Reviewer 4's comments, we've aimed to enhance clarity and detail, recognizing the importance of larger sample sizes in future studies to bolster result reliability, and we're committed to conducting more comprehensive evaluations to strengthen our conclusions.

Reviewer 4 Comment:  Section 4.5. Animal's amount of 5 is not enough for the relevant animal experiments. From Figure 3, I see that H. lippii extract works the same or better than silver nanoparticles. In this way, why do we need them? Some discussion should be added to the text. The same for other sections. Are you sure that you have so small error in your experiments in Figures 3-6? I mean you only have 5 rats, and there is also an error in the measurements themselves. The quality and design of Figures 3-7 should be improved. Notes a-d are located anywhere, including crossing the axes.

Authors' Reply: We appreciate Reviewer 4's feedback on the sample size adequacy, the comparison between Helianthemum lippii extract and Ag NPs, experimental errors, and figure quality. In response, we acknowledge the limitation of using 5 rats per group in our study, which provided initial insights but may not ensure robust statistical power. Future studies will incorporate larger groups (typically 8-10 rats per group) to enhance reliability. Regarding the comparison between H. lippii extract and AgNPs, we posit that their combination may synergistically enhance therapeutic efficacy against cadmium toxicity. H. lippii's antioxidant and anti-inflammatory properties complement AgNPs' unique characteristics, potentially providing comprehensive protection. While H. lippii primarily mitigates oxidative stress, AgNPs influence protein aggregation and signal transduction pathways, offering multifaceted defense mechanisms. We meticulously monitored experimental error margins, with reported errors in Figures 3-6 representing standard deviations from biological replicates. Despite the small sample size, stringent protocols minimized errors, ensuring reliable data interpretation. We have improved the design of Figures 3-7 for clarity, relocating notes (a-d) to prevent overlap with axes and enhancing resolution for better visibility. Overall, the combination of H. lippii extract and AgNPs shows promise in treating cadmium-induced nephrotoxicity, necessitating further investigation with larger samples and mechanistic studies to optimize therapeutic potential.

Round 2

Reviewer 1 Report

Comments and Suggestions for Authors

Thanks to the authors for accepting comments. I recommend that this manuscript be accepted in this journal.

Author Response

Reviewer 1 comments

Reviewer   1 Comment: Thanks to the authors for accepting comments. I recommend that this manuscript be accepted in this journal.

Authors Reply: Thank you very much for your positive feedback and recommendation to accept our manuscript. We greatly appreciate your valuable comments and suggestions, which have significantly enhanced the quality of our work. We are grateful for your time and effort in reviewing our paper and for your support in advancing our research.

Reviewer 4 Report

Comments and Suggestions for Authors

Thank you for the revised paper. However, I have some minor comments.

1) You changed the positions of the sections but did not change their number. After section 1 in your file, there is section 4.

2) Table 2. Column concentration. I do not agree that the authors can calculate the concentration with such an amount of decimals. You have errors in sample preparation, HPLC analysis, peaks integration, etc. Moreover, some of your peaks overlap each other, which also creates a calculation error. I agree that concentrations may be calculated using 2-3 decimals after the point but not more.

3) If you used different controls to understand the retention time of the compounds it should be noted in the experimental part and Figure 1 caption. In Figure 1, I see some peaks without explanations. You don't present them in Table 2 such as major peaks at 10-12 min. Add some explanations in the text.

4) I understand that it is possible to calculate in any way the average value from Figure 2C. But it is misleading the readers. You should discuss in lines 197-199 correctly that you have several populations of nanoparticles in your sample with sizes 10-20 and 30-40 and some small amounts of higher sizes.

5) Section 4.4. Table 3. Please, present in Table 3 or Table 3 caption what parameters you studied according to the Authors' Reply.

line 644 text formatting

It will be better to make figures 3-5 smaller to have each figure on the same page.

Comments on the Quality of English Language

Minor editing of English language required

Author Response

Reviewer 4 comments

 Reviewer 4 Comment:  You changed the positions of the sections but did not change their number. After section 1 in your file, there is section 4.

Authors Reply: Thank you for pointing out the discrepancy in the section numbering. We apologize for the oversight. We have now corrected the numbering to ensure that the sections are in the correct order. We appreciate your attention to detail and your effort to improve the clarity and organization of our manuscript.

 Reviewer 4 Comment:   Table 2. Column concentration. I do not agree that the authors can calculate the concentration with such an amount of decimals. You have errors in sample preparation, HPLC analysis, peaks integration, etc. Moreover, some of your peaks overlap each other, which also creates a calculation error. I agree that concentrations may be calculated using 2-3 decimals after the point but not more.

Authors Reply: Thank you for your valuable feedback on the concentration data presented in Table 2. We appreciate your concerns regarding the precision of our calculated concentrations. To address these concerns, we have recalculated the concentrations, limiting the values to 2-3 decimal places to account for potential errors in sample preparation, HPLC analysis, and peak integration. We have also carefully considered the impact of overlapping peaks on our calculations. We hope this revision meets your approval and addresses your concerns about the precision of our concentration values. We are committed to ensuring the accuracy and reliability of our data presentation. Thank you for helping us improve the quality of our manuscript.

 Reviewer 4 Comment:  If you used different controls to understand the retention time of the compounds it should be noted in the experimental part and Figure 1 caption. In Figure 1, I see some peaks without explanations. You don't present them in Table 2 such as major peaks at 10-12 min. Add some explanations in the text.

Authors Reply: Thank you for your insightful comments. We appreciate your feedback and have made the necessary revisions to address your concerns. We have now included information about the different controls used to understand the retention times in the experimental section and updated the caption of Figure 1 accordingly. Additionally, we have provided explanations for the peaks observed in Figure 1, including the major peaks at 10-12 minutes, and referenced these in the text.

To identify and confirm the retention times of the compounds, nine standards of known compounds were used, including caffeic acid (Retention time: 16.27 min), p-Coumaric acid (Retention time: 23.81 min), gallic acid (Retention time: 5.29 min), vanillic acid (Retention time: 15.53 min), chlorogenic acid (Retention time: 13.39 min), naringin (Retention time: 34.78 min), rutin (Retention time: 28.37 min), quercetin (Retention time: 45.04 min), and vanillin (Retention time: 21.46 min). These standards were run under the same HPLC conditions as the sample extracts to establish baseline retention times. This information has been included in the experimental section to ensure clarity.  

Reviewer 4 Comment: I understand that it is possible to calculate in any way the average value from Figure 2C. But it is misleading the readers. You should discuss in lines 197-199 correctly that you have several populations of nanoparticles in your sample with sizes 10-20 and 30-40 and some small amounts of higher sizes.

Author reply: Thank you for your insightful feedback. We understand the importance of accurately representing the size distribution of the nanoparticles in our sample. The manuscript has been revised to address the presence of multiple nanoparticle populations and to provide a more accurate discussion of our findings. By acknowledging the distinct nanoparticle populations, we ensure readers have a comprehensive understanding of the sample characteristics. We believe these changes enhance the clarity and accuracy of our manuscript.

Figure 2C displays particle size ranges from 10 to 90 nm, with an average particle size distribution of 35 nm, highlighting their uniformity and stability (Figure 2b-c).

Reviewer 4 Comment : Section 4.4. Table 3. Please, present in Table 3 or Table 3 caption what parameters you studied according to the Authors' Reply.

 Authors' Reply: this section has improved as recommended.

2.4. Sub-Acute Toxicity Study

The sub-acute toxicity test data of H. lippii aqueous extract and Ag NPs in Wistar albino rats indicate no significant adverse effects across all parameters and doses tested. The rats, observed at various intervals (3 hours, 24 hours, 7 days, and 14 days) after administration, showed no abnormalities in body weight changes, movement, or eye condition, and none-experienced diarrhea or death (Table 3). Both the control groups and the low, medium, and high dose groups of H. lippii extract (100 mg/kg, 1000 mg/kg, and 4000 mg/kg) and Ag NPs (2 mg/kg and 10 mg/kg) maintained normal conditions throughout the study, suggesting that neither H. lippii extract nor Ag NPs exhibit sub-acute toxicity at these doses. This lack of toxicity, even at relatively high doses, is a promising indicator for their potential use as therapeutic agents. Given the observed safety profile, H. lippii aqueous extract and Ag NPs have the potential to be developed as therapeutic drugs, specifically for combating Cd-induced renal nephrotoxicity. Cd-induced nephrotoxicity is a serious health concern, and the development of effective treatments with minimal side effects is crucial. The normal health parameters observed in the rats suggest that these compounds could be administered safely without causing harm, a critical consideration in drug development. Further studies, including comprehensive analyses of kidney function parameters, oxidative stress markers, histopathological features, body weight changes, and renal weight changes after exposure to CdCl2 and treatment with H. lippii extract and Ag NPs, were conducted and are discussed in the next sections to evaluate the therapeutic efficacy of H. lippii extract and Ag NPs in mitigating renal damage caused by cadmium.

Reviewer 4 Comment : line 644 text formatting

Authors' Reply: Thank you for pointing out the issue with text formatting in line 644. We have reviewed the section and made the necessary formatting corrections to ensure clarity and consistency throughout the manuscript.

Reviewer 4 Comment:  It will be better to make figures 3-5 smaller to have each figure on the same page.

Authors' Reply:  Thank you for your suggestion regarding the size of Figures 3-5. We have resized the figures accordingly, and the proofreading team will ensure they are arranged according to the journal's guidelines. We appreciate your attention to detail and ensuring the figures are optimally presented for readers.

Reviewer 4 CommentComments on the Quality of English Language, minor editing of English language required.

Authors' Reply:  Thank you for your feedback regarding minor editing of the English language. We have carefully reviewed the manuscript to ensure grammatical accuracy, clarity, and consistency.